



# Automated quantification of floating wood pieces in rivers
# from video monitoring: a new software tool and validation.
Hossein Ghaffarian[1, *,] Pierre Lemaire[1,2], Zhang Zhi[1], Laure Tougne[2], Bruce MacVicar[3], and Hervé Piégay[1]
[1]Univ. Lyon, UMR 5600, Environnement-Ville-Société CNRS, F-69362 Lyon, France
[2]Univ. Lyon, UMR 5205, Laboratoire d'InfoRmatique en Image et Systèmes d'information CNRS, F-69676
Lyon, France
3Department of Civil and Environmental Engineering, Univ. Waterloo, Waterloo, Ontario, Canada
*Correspondence to*: Hossein Ghaffarian (hossein.ghaffarian@ens-lyon.fr)
**Abstract**
Wood is an essential component of rivers and plays a significant role in ecology and morphology. It can
be also considered as a risk factor in rivers due to its influence on erosion and flooding. Quantifying and
characterizing wood fluxes in rivers during floods would improve our understanding of the key processes but
is hindered by technical challenges. Among various techniques for monitoring wood in rivers, streamside
videography is a powerful approach to quantify different characteristics of wood in rivers, but past research
has employed a manual approach that has many limitations. In this work, we introduce new software for the
automatic detection of wood pieces in rivers. We apply different image analysis techniques such as static and
dynamic masks, object tracking, and object characterization to minimize false positive and missed detections.
To assess the software performance, results are compared with manual detections of wood from the same
videos, which was a time-consuming process. Key parameters that affect detection are assessed including
surface reflections, lighting conditions, flow discharge, wood position relative to the camera, and the length
of wood pieces. Preliminary results had a 36% rate of false positive detection, primarily due to light reflection
and water waves, but post-processing reduced this rate to 15%. The missed detection rate was 71% of piece
numbers in the preliminary result, but post processing reduced this error to only 6.5% of piece numbers, and
13.5% of volume. The high precision of the software shows that it can be used to massively increase the
quantity of wood flux data in rivers around the world, potentially in real time. The significant impact of post-
processing indicates that it is necessary to train the software in various situations (location, timespan, weather
conditions) to ensure reliable results. Manual wood detections and annotations for this work took more than
one human-month of labor. In comparison, the presented software coupled with an appropriate post pro-
cessing step performed the same task in real time (55 hr) on a standard desktop computer.





Keywords: River monitoring, Wood flux, Wood discharge, Large wood, Ground video imagery, Auto-
matic detection

## 1.    Introduction

Floating wood has a significant impact on river morphology (Gurnell et al., 2002; Gregory et al., 2003;
Wohl, 2013; Wohl and Scott, 2017). It is both a component of stream ecosystems and a source of risk for
human activities (Comiti et al., 2006; Badoux et al., 2014; Lucía et al., 2015). The deposition of wood at
given locations can cause a reduction of the cross-sectional area, which can both increase upstream water
levels (and the risk for neighboring communities), and laterally concentrate the flow downstream, which can
lead to damaged infrastructure (Lyn et al., 2003; Lagasse, 2010; Mao and Comiti, 2010; Badoux et al., 2014;
Ruiz-Villanueva et al., 2014; De Cicco et al., 2018; Mazzorana et al., 2018). Therefore, understanding and
monitoring the dynamics of wood within a river is fundamental to assess and mitigate risk. An important
body of work on this topic has grown over the last two decades, which has led to the development of many
monitoring techniques (Marcus et al., 2002; MacVicar et al., 2009a; MacVicar and Piégay, 2012; Benacchio
et al., 2015; Ravazzolo et al., 2015; Ruiz-Villanueva et al., 2018; Ghaffarian et al., 2020; Zhang et al., 2020)
and conceptual and quantitative models (Braudrick and Grant, 2000; Martin and Benda, 2001; Abbe and
Montgomery, 2003; Gregory et al., 2003; Seo and Nakamura, 2009; Seo et al., 2010). A recent review by
Ruiz-Villanueva et al. (2016), however, argues that the area remains in relative infancy compared to other
river processes such as the characterization of channel hydraulics and sediment transport. Many questions
remain open areas of inquiry including wood hydraulics, which is needed to understand wood recruitment,
movement and trapping, and wood budgeting, where better parametrization is needed to understand and
model the transfer of wood in watersheds at different scales.
In this domain, the quantification of wood mobility and wood fluxes in real rivers is a fundamental
limitation that constrains model development. Most early works were based on repeated field surveys (Keller
and Swanson, 1979; Lienkaemper and Swanson, 1987), with more recent efforts taking advantage of aerial
photos or satellite images (Marcus et al., 2003; Lejot et al., 2007; Lassettre et al., 2008; Senter and Pasternack,
2011; Boivin et al., 2017) to estimate wood delivery at larger time scales of 1 year up to several decades.
Others have monitored wood mobility once introduced by tracking wood movement in floods (Jacobson et
al., 1999; Haga et al., 2002; Warren and Kraft, 2008). Tracking technologies such as active and passive Radio
Frequency Identification transponders (MacVicar et al., 2009a; Schenk et al., 2014) or GPS emitters and



receivers (Ravazzolo et al., 2015) can improve the precision of this strategy. To better understand wood flux,
specific trapping structures such as reservoirs or hydropower dams can be used to sample the flux over time
interval windows (Moulin and Piégay, 2004; Seo et al., 2008; Turowski et al., 2013). Accumulations up-
stream of a retention structure can also be monitored where they trap most or all of the transported wood, as
was observed by Boivin *et al.* (2015), to quantify wood flux at the flood event or annual scale. All these
approaches allow the assessment of wood budget and the in-channel wood exchange between geographical
compartments within a given river reach and over a given period (Schenk et al., 2014; Boivin et al., 2015,
2017).

For finer scale information on the transport of wood during flood events, video recording of the water
surface is suitable for estimating instantaneous fluxes and size distributions of floating wood in transport
(Ghaffarian et al., 2020). Classic monitoring cameras installed on the river bank are cheap and relatively easy
to acquire, setup and maintain. As is seen in Table 1, a wide range of sampling rates and spatial/temporal
scales have been used to assess wood budgets in rivers. MacVicar and Piégay (2012) and Zhang et al., (2020)
(in review), for instance, monitored wood fluxes at 5 frames per second (fps) and a resolution of $640 \times 480$
up to $800 \times 600$ pixels. Boivin et al. (2017) used a similar camera and frame rate as MacVicar and Piégay
(2012) to compare periods of wood transport with and without the presence of ice. Senter et al. (2017) ana-
lyzed the complete daytime record of 39 days of videos recorded at 4 fps and a resolution of $2048 \times 1536$
pixels. Conceptually similar to the video technique, time-lapse imagery can be substituted when large rivers
where surface velocities are low enough and the field of view is large. Kramer and Wohl (2014); Kramer et
al. (2017) applied this technique in the Slave River (Canada) and recorded one image every 1 and 10 minutes.
Where possible, wood pieces within the field of view are then visually detected and measured using simple
software to measure the length and diameter of the wood to estimate wood flux (piece/s) or wood volume
($m^3/s$) (MacVicar and Piégay, 2012; Senter et al., 2017). Critically for this approach, the time it takes for
the researchers to extract information about wood fluxes has limited the fraction of the time that can be
reasonably analyzed. Given the outdoor location for the camera, the image properties depend heavily on
lighting conditions (e.g. surface light reflections, low light, ice, poor resolution or surface waves) which may
also limit the accuracy of frequency and size information (Muste et al., 2008; MacVicar et al., 2009a). In
such situations, simpler metrics such as a count of wood pieces, a classification of wood transport intensity,
or even just a binary presence/absence may be used to characterize the wood flux (Boivin et al., 2017; Kramer
et al., 2017).



**Table 1**

A fully automatic wood detection and characterization algorithm can greatly improve our ability to
exploit the vast amounts of data on wood transport that can be collected from streamside video cameras.
From a computer science perspective, however, automatic detection and characterization remain challenging
issues. In computer vision, detecting objects within videos typically consists of separating the foreground
(the object of interest) from the background (Roussillon et al., 2009; Cerutti et al., 2011, 2013). The basic
hypothesis is that the background is relatively static and covers a large part of the image, allowing it to be
matched between successive images. In the riverine environments, however, such an assumption is unrealistic
because the background shows a flowing river, which can have rapidly fluctuating properties (Ali and
Tougne, 2009). Floating objects are also partially submerged in water that has high suspended material con-
centrations during floods, making them only partially visible (*e.g.* a single piece of wood may be perceived
as multiple objects) (MacVicar et al., 2009b). Detecting such an object in motion within a dynamic back-
ground is an area of active research (Ali et al., 2012, 2014; Lemaire et al., 2014; Piégay et al., 2014; Be-
nacchio et al., 2017). Accurate object detection typically relies on the assumption that objects of a single
class (e.g. faces, bicycles, animals, etc.) have a distinctive aspect or set of features that can be used to distin-
guish between types of objects. With the help of a representative dataset, machine learning algorithms aim
at defining the most salient visual characteristics of the class of interest (Lemaire et al., 2014; Viola and
Jones, 2006). When the objects have a wide intra-class aspect range, a large amount of data can compensate
by allowing the application of deep learning algorithms (Gordo et al., 2016; Liu et al., 2020). To our
knowledge, such a database is not available in the case of floating wood.
The camera installed on the Ain River in France has been operating more or less continuously for over
10 years and vast improvements in data storage mean that this data can be saved indefinitely (Zhang et al.,
2020). The ability to process this image database to extract the wood fluxes allows us to integrate this infor-
mation over floods, seasons and years, which would allow us to significantly advance our understanding of
the variability within and between floods over a long time period. An unsupervised method to identify float-
ing wood in these videos by applying intensity, gradient and temporal masks was developed by Ali and
Tougne (2009) and Ali *et al.* (2011). In this model, the objects were tracked through the frame to ensure that
they followed the direction of flow. An analysis of about 35 minutes of the video showed that approximately
90% of the wood pieces was detected (*i.e.* about 10% of detection were missed), which confirmed the poten-
tial utility of this approach. An additional set of false detection related to surface wave conditions amounted
to approximately 15% of the total detection. However, the developed algorithm was not always stable and



was found to perform poorly when applied to a larger data set.
The objectives of the presented work are to describe and validate a new algorithm and computer inter-
face for quantifying floating wood pieces in rivers. First, the algorithm procedure is introduced to show how
wood pieces are detected and characterized. Second, the computer interface is presented to show how manual
annotation is integrated with the algorithm to train the detection procedure.  Third, the procedure is validated
using data from the Ain River. The validation period occurred over six days in January and December 2012
where flow conditions ranged from $\sim 400 \ m^3/s$, which is below bankfull discharge but above the wood
transport threshold, to more than $800 \ m^3/s$. The developed algorithm can be used to characterize wood
pieces for a large image database at the study site.  Future applications of this approach at a wide range of
sites should lead to new insights on the variability of wood pieces at the reach and watershed scales in world
rivers.
**2.      Methodological procedure for automatic detection of wood**
The algorithm for wood detection comprises a number of steps that seek to locate objects moving
through the field of view in a series of images and then identify the objects most likely to be wood. The
algorithm used in this work modifies the approach described by Ali *et al.*, (2011). The steps work from a
pixel to image to video scale, with the context from the larger scale helping to assess whether the information
at the smaller scale indicates the presence of floating wood or not. In a still image, a single pixel is charac-
terized by its location within the image, its color and its intensity. Looking at its surrounding pixels, on an
image scale, allows that information to be spatially contextualized. Meanwhile, the video data adds temporal
context, so that previous and future states of a given pixel can be used to assess its likeliness of representing
floating wood. Since an image is only a discrete 2D representation of the real 3D world, details about the
camera parameters such as optical image deformations, geographic situation, perspective deformations or
behavior regarding luminosity can be used to infer what wood should look like and where it should occur.
On a video scale, the method can embed expectations about how wood pieces should move through frames,
how big they should be, and how lighting and weather conditions can evolve to change the expectations of
wood appearance, location, and movement. The specific steps followed by the algorithm are shown in a
simple flow chart (Fig 1.a). An example image with a wood piece in the middle of the frame is also shown
for reference (Fig 1.b).

**Fig 1**



## 2.1. Wood probability masks


In the first step, each pixel was analyzed individually and independently. The static probability mask
answers the question "is one pixel likely to belong to a wood-block, given its color and intensity?". The
algorithm assumes that the wood pixels can be identified by pixel light intensity ($x$) following a Gaussian
distribution (Fig 2.a). To set the algorithm parameters, manual annotations of wood are used to obtain a
representative sample of wood pixels, from which both the mean ($\mu$) and standard deviation ($\sigma$) are calcu-
lated. This procedure produces a static probability mask (Fig 2.b). From this figure, it is possible to identify
the sectors where wood presence is likely, which includes the floating wood piece seen in Fig 1.b, but also
includes standing vegetation in the lower part of the image and a shadowed area in the upper left. The ad-
vantage of this approach is that it is computationally very fast. However, misclassification is possible, par-
ticularly when light condition changes.

**Fig 2**

The second mask, called the dynamic probability mask, outlines each pixel's recent history. The corre-
sponding question is: "is this pixel likely to represent wood now, given its past and present characteristics?".
Again, this step is based on what is most common in our database: it is assumed that a wood pixel is darker
than a water pixel. Depending on lighting conditions like shadows cast on water or waves, this is not always
true, i.e. water pixels can be as dark as wood pixels. However, pixels displaying successively water then
wood tend to become immediately and significantly darker, while pixels displaying wood then water tend to
become significantly lighter. Meanwhile, pixels that keep on displaying wood tend to be rather stable. Thus,
we assign wood pixel probability according to an updated version of the function proposed by Ali et al.
(2011) (Fig 3.a) that takes 4 parameters. This function $H$ is an updating function, which produces a temporal
probability mask from the inter-frame pixel value. On a probability map, a pixel value ranges from -1 (likely
not wood) to 1 (likely wood). The temporal mask value for a pixel at location $(x, y)$ and at time $t$ is
$P_T(x, y, t) = H(\Delta_t, I) + P_T(x, y, t - 1)$. We apply a threshold to the output of $P_T(x, y, t)$ so that it always
stays within the interval [0,1]. The idea is that a pixel that becomes suddenly and significantly darker is
assumed to be likely wood. $H(\Delta_t, I)$ is such that under those conditions, it increases the pixel probability map
value (parameters $\tau$ and $\beta$). A pixel that becomes lighter over time is unlikely to correspond to wood (pa-
rameter $\alpha$). A pixel which intensity is stable and that was previously assumed to be wood shall still corre-
spond to wood, while a pixel which intensity is stable and which probability to be wood was low is unlikely
to represent wood now. A small decay factor ($\delta$) was introduced in order to prevent divergence (in particular,





it prevents noisy areas from being activated too frequently).

**Fig 3**

The final wood probability mask is created using a combination of both the static and dynamic proba-
bility masks. Wood objects thus had to have a combination of the correct pixel color and the expected tem-
poral behavior of water-wood-water color. The masks were combined assuming that both probabilities are
independent, which allowed us to use the Bayesian probability rule in which the probability masks are simply
multiplied, pixel by pixel, to obtain the final probability value for each pixel of every frame.

## 2.2. Wood object identification and characterization

From the probability mask it is necessary to group pixels with high wood probabilities into objects and
then to separate these objects from the background to track them through the image frame. For this purpose,
pixels were classified as high-or low-probability based on a threshold applied to the combined probability
mask. Then, the high-probability pixels were grouped into connected components (that is, small, contiguous
regions on the image) to define the objects. At this stage, a pixel size threshold was applied on the detected
objects so that only the bigger objects were considered to represent woody objects on the water surface (Fig
4.a the big white region at the middle). A number of smaller components were often related to non-wood
objects, for example waves, reflections, or noise from the camera sensor or data compression.
After the size thresholding step, movement direction and velocity were used as filters to distinguish real
objects from false detections. The question here is, "is this object moving through the image frame the way
we would expect floating wood to move?". To do this, the spatial and temporal behavior of components were
analyzed. First, to deal with partly immersed objects, we agglomerated multiple objects within frames as
components of a single object if the distance separating them was less than a set threshold. Second, we asso-
ciated wood objects in successive frames together to determine if the motion of a given object was compatible
with what is expected from driftwood. This can be achieved according to the dimensionless parameter
"$PT/\Delta T$", which provides a general guideline for the distance an object pass between two consecutive frames
(Zhang et al., 2020). Here $PT$ (passing time) is the time that one piece of wood passes through the camera
field of view and $\Delta T$ is the time between two consecutive frames and practically it is recommended to use
videos with $PT/\Delta T > 5$ in this software. In our case, tracking wood is rather difficult for classical object
tracking approaches in computer vision: the background is very noisy, the acquisition frequency is low and
the objects appearance can be highly variable due to temporarily submerged parts and highly variable 3D



structures. Given these considerations it was necessary to use very basic rules for this step. The rules are
therefore based on loose expectations, in terms of pixel intervals, on the motions of the objects, depending
on the camera location and the river properties.  How many pixels is the object likely to move between image
frames from left to right? How many pixels from top to bottom? How many appearances are required? How
many frames can we miss because of temporary immersions? Using these rules, computational costs re-
mained low and the analysis could by run in real-time while also providing good performance.

**Fig 4**

The final step was to characterize each object, which at this point in the process are considered wood
objects. Each appears several times in different frames and a procedure is needed to either pick a single
representative occurrence or use a statistic tool to analyse multiple occurrences to estimate characterization
data. Here we assumed that the biggest occurrence, in terms of pixels number, was the most representative
state. This assumption is based on the principle that a bigger number of pixels corresponds to a better or a
fuller view (the object is less immersed than on other occurrences, for instance). This approach also matched
the manual annotation procedure where we tended to pick the view where the object covers the largest area
to make measurements. For the current paper, every object as characterized from the raw image based on its
size and its location (in pixels).

### 2.3. Image rectification

Warping images according to a perspective transform results in an important loss of quality. On warped
images, areas of the image farther to the camera provide little detail and are overall very blurry and non-
informative. Therefore, given the topology of our images, image rectification was necessary to calculated
wood length, velocity, volume and other characteristics from the saved pixel-based characterization of each
object. To do so, the fisheye lens distortion was first corrected. A fisheye lens distortion is a characteristic of
the lens that produces visual distortion intended to create a wide panoramic or hemispherical image. This
effect was corrected by a standard Matlab process using the ComputerVisionToolbox$^{TM}$.
Ground-based cameras have also an oblique angle of view, which means that pixel to meter correspond-
ence is variable and images need to be orthorectified to obtain estimates of object size and velocity in real
terms (Muste et al., 2008). Orthorectification refers to the process by which image distortion is removed and
the image scale is adjusted to match the actual scale of the water surface. Translating from pixels to cartesian
coordinates required us to assume that our camera follows the pinhole camera model and that the river can





be assimilated to a plane of constant altitude. Under such conditions, it is possible to translate from pixel
coordinates to a metric 2D space thanks to a perspective transform assuming a virtual pinhole camera on the
image and estimating the position of the camera and its principal point (center of the view). An example of
orthorectification on a detected wood piece in a set of continuous frames and pixel coordinates (Fig 5.b) is
presented in Fig 5.c in metrics coordinates. The transform matrix is obtained with the help of at least 4 non-
colinear points (Fig 5.a blue GCPs (Ground Control Points) acquired with DGPS) from which we know both
the relative 2D metric coordinates for a given water level (Fig 5.c blue points), and their corresponding lo-
calization within the image(Fig 5.b blue points). To achieve better accuracy, it is advised to acquire additional
points and to solve the subsequent over-determined system with the help of a Least Square Regression (LSR).
Robust estimators such as RANSAC can provide useful to prevent acquisition noise. After identifying the
virtual camera position, the perspective transform matrix then becomes parameterized with the water level.
Handling the variable water level was performed for each piece of wood, by measuring the relative height
between the camera and the water level at the time of detection based on information recorded at the gauging
station to which the camera was attached.

**Fig 5**

**3.    User interface**
The software was developed to provide a single environment for the analysis of wood pieces on the
surface of the water from streamside videos.  It consists of four distinct modules: Detection, Annotation,
Learning, and Performance. The home screen (Fig 6) allows the operator to select any of these modules.
From within a module, a menu bar on the left side of the interface allows operators to switch from one module
to another. In the following sections, the operation of each of these modules are described.

**Fig 6**

**3.1.  Detection**
The detection module is the heart of the software. This module allows, from learned or manually spec-
ified parameters, the detecting of floating objects without human intervention (see Fig 7). This module con-
tains two main parts: (i) Detection tab, which allows operator to open, analyze and export the results from
one video or a set of videos, and (ii) Configuration tab, which allows operator to load and save the software
configuration by defining the parameters of wood detection (as described in Sect 2), saving and extracting
the results, and displaying the interface.





The detection process is intended to work as a video file player. The idea is to load a video file (or a

stream url), and to let the software read the video until the end. When required, the reader generates a visual
output, showing how the masks behave by adding color and information to the video content (see Figure 7).
A small textual display area shows the frequency of past detections. Meanwhile, the software generates a
series of files summarizing the positive outputs of the detection. They consist in YAML and CSV files, as
well as image files to show the output of different masks, the original frames, etc. A configuration tab is
available, and provides many parameters organized by various categories. The main configuration tab is
divided in seven parts. The first part is dedicated to general configurations such as frame skipped between
each computation and defining the areas within the frame where wood is not expected (e.g. bridge pier or
river bank). In the second and third parts, the parameters of the intensity and temporal masks are listed (see
Sect 2.1). The default values are $\mu = 0.2$ and $\sigma = 0.08$ for the intensity mask, and $\tau = 0.25$ and $\beta = 0.45$
for the temporal mask. In the fourth and fifth parts, object tracking and characterization parameters are de-
fined respectively as described in Sect 2.2. Detection time is defined in the sixth part using an optical char-
acter recognition technique. Finally, the parameters of the orthorectification (see Sect 2.3) are defined in the
seventh part. The detection software can be used to process videos in batch ("script" tab), without generating
a visual output to save computing resources.

**Fig 7**

**3.2. Annotation**

As mentioned in Sec. 2, the detection procedure requires the classification of pixels and objects into

wood and non-wood categories. To train and validate the automatic detection process, a ground-truth or set
of videos with manually annotations are required. Such annotations can be performed using different tech-
niques. For example, objects can be identified with the help of a bounding box or selection of endpoints, as
in MacVicar and Piégay(2012); Ghaffarian et al., (2020) and Zhang et al., (2020). It is also possible to sample
wood pixels without specifying instances or objects, or to sample pixels within annotated objects. Finally,
objects and/or pixels can be annotated multiple times in a video sequence to increase the amount and detail
of information in such an annotation database. However, this annotation process is time-consuming, so a
trade-off must be made between training and accuracy for different lighting conditions, camera parameters,
wood properties, and river hydraulics.

Given that the tool is meant to be as flexible as possible, the annotation tool was developed to allow

operator to perform as fine annotation as they wish. As it is shown in Fig 8, this module contains three main





parts: (i) The column on the far left allows operator to switch to another module (detection, learning or per-
formance), (ii) the central part consists of a video player with a configuration tab for extracting the data, and
(iii) the right part where the tools to generate, create, visualize and save annotations are located. The tools
allow rather quick coarse annotation, similar to what was done by MacVicar and Piégay (2012) and Boivin
*et al.,* (2015), while still allowing the possibility of finer pixel-scale annotation.

**Fig 8**

The principle of this module is to associate annotations with the frames of a given video. Annotating a
piece of wood is like drawing its shape, directly on a frame of the video, using the drawing tools provided by
the module. It is possible to add a text description to each annotation. Each annotation is linked to a single
frame of the video; however, a frame can contain several annotations. An annotated video, therefore, consists
of a video file, as well as a collection of drawings, possibly with textual descriptions, associated with frames.
It is possible to link annotations from one frame to another to signify that they belong to the same piece of
wood. These data can be used to learn the movement of pieces of wood in the frame.
**3.3. Performance**
The performance module allows the operator to set rules to compare automatic and manual wood de-
tection results. This section also allows the operator to use a bare, pixel-based annotation or specify an or-
thorectification matrix to extract wood-size metrics directly from the output of an automatic detection.
For this module an automatic detection file is first loaded and then the result of this detection is com-
pared with a manual annotation for that video, if the latter is available. Comparison results are then saved in
the form of a summary file (*.csv format), allowing the operator to perform statistical analysis of the results
or the performance of the detection algorithm. A manual annotation file can only be loaded if it is associated
with an automatic detection result.
The performance of the detected algorithm can be realized on several levels:
•     Object. The idea is to annotate one (or more) occurrences of a single object, and to operate the

comparison at bounding box scale. A detected object may comprehend a whole sequence of occur-

rences, on several frames. It is validated when only a single occurrence happens to be related to an

annotation. This is the minimum possible effort required to have an extensive overview of the

object frequency on such an annotations database. This approach can however lead us to misjudge



overall wrongly detected sequences as True Positives (see below), or vice-versa.

•   Occurrence. The idea is to annotate, even roughly, every occurrence of every woody object, so that

the comparison can happen between bounding boxes rather than at pixel level. Every occurrence

of any detected object can be validated individually. This option requires substantially more anno-

tation work than the object annotation.

•   Pixel. This case implies that every pixel of every occurrence of every object is annotated as wood.

It is very powerful in the event of evaluating the algorithm performances, and eventually refining

its parameters with the help of some machine learning technique. However, it requires an extensive

annotation work.

### 324     4.    Performance assessment

To assess the performance of the automatic detection algorithm, we used a set of videos from the Ain

River in France that were both comprehensively manually annotated and automatically analyzed. According
to the data annotated by the observer, the performance of the software can be affected by different conditions:
(i) wood piece length, (ii) distance from the camera, (iii, iv) wood X, Y position, (v) flow discharge, (vi)
daylight, and (vii, viii) light and darkness of the frame (see Table 2). If for example software detects a 1 cm
piece at a distance of 100 m from the camera, there is a high probability that this is a false positive detection.
Therefore, knowing the performance of the software in different conditions, it is possible to develop some
rules to enhance the quality of data.  The advantage of this approach is that all eight parameters introduced
here are accessible easily in the detection process. In this section the monitoring details and annotation meth-
ods are introduced before the performance of the software is evaluated by comparing the manual annotations
with the automatic detections.

**Table 2**

### 336    4.1.  Material and methods

### 337    4.1.1.  Monitoring site and annotation

The Ain River is a piedmont river with a drainage area of 3630 $km^2$ at the gauging station of Chazey-

sur-Ain, with a mean flow width of 65 m, a mean slope of 0.15%, and a mean annual discharge of 120 $m^3/s$.
The lower Ain River is characterized by an active channel shifting within a forested floodplain (Lassettre et
al., 2008). An AXIS221 Day/Night$^{TM}$ camera with a resolution of $768 \times 576$ pixels was installed at this station





to continuously record the water surface of the river at a maximum frequency of 5 fps (Fig 9). This camera
replaced a lower resolution camera at the same location used by MacVicar and Piégay (2012). The specific
location of the camera is on the outer bank of a meander, on the side closest to the thalweg, at a height of 9.8
m above the base flow elevation. The meander and a bridge pier upstream help to steer most of the floating
wood so that it passes relatively close to the camera where it can be readily detected with a manual procedure
(MacVicar and Piégay, 2012). The transformation matrix at the base flow elevation with the camera as the
origin is shown in Fig 10. Straight lines near the edges of the image appear curved because the fisheye
distortion has been corrected on this image (see Sect 2.3); conversely, a straight line, in reality, is presented
without any curvature in the image.

**Fig 9**

**Fig 10**

The survey period examined on this river was during 2012 from which two flood events, (January 1-7
and December 15) were selected for annotation. A range of discharges from $400 m^3/s$ to $800\ m^3/s$ occurred
during these periods (Fig 11), which is above a previously observed wood transport threshold of $\sim 300\ m^3/s$
(MacVicar and Piégay, 2012). The flow discharge is available from the website (*www.hydro.eaufrance.fr*).
On January 3$^{rd}$ and 5$^{th}$, a spider was active in front of the camera, which prevented a good video recording
and these days were therefore removed from the database. Detection was only possible during the daylight.
A summary of automated and manual detections for the six days is shown in Table 3.

**Fig 11**

**4.1.2.  Assessment Methodology**

Ghaffarian et al. (2020), Zhang et al. (2020) show that the wood discharge can be measured from flux
or frequency of wood objects. An object level detection was thus sufficient for the larger goals of this re-
search at the Ain River, which is to get a complete budget of transported wood volume.
A comparison of annotated with automatic object detections gives rise to three options:
1- True Positive ($TP$): an object is correctly detected and is recorded in both the automatic and annotated

database

2- False Positive ($FP$): an object is incorrectly detected and is recorded only in the automatic database.
3- False Negative ($FN$): an object is not detected automatically and is only recorded in the annotated

Earth **Surface**
**Dynamics**
Discussions

database.
Despite overlapping occurrences of wood objects in the two databases, the objects could vary in position
and size between them. For the current study we set the TP threshold as the case where either at least 50%
of the automatic and annotated bounding box areas were common or at least 90% of an automatic bounding
box area was part of its annotated counterpart.
In addition to the raw counts of $TPs$, $FPs$, and $FNs$, we defined two measures of the performances of
the application, where:
• Recall Rate ($RR$) is the fraction of wood objects that are automatically detected ($TP/(TP + FN)$); and
• Precision Rate ($PR$) is the fraction of detected objects that are wood ($TP/(TP + FP)$).
The higher the $PR$ and the $RR$ are, the more accurate our application is. However, both rates tend to
interact. For example, it is possible to design an application that displays a very high $RR$ (which means that
it doesn't miss many objects), but suffers from a very low $PR$ (it outputs a high amount of inaccurate data),
and vice-versa. Thus, we have to find a balance that is appropriate to each application.
**4.1.3. Factors used to understand variation in performance**
It was well known from previous manual efforts to characterize wood pieces and develop automated
detection tools that it is easier to detect certain wood objects than others. In general, the ability to detect the
wood objects in the dynamic background of a river in flood was found to vary with the size of the wood
object, its position in the image frame, the flow discharge, the amount and variability of the light, interference
from other moving objects such as spiders, and other weather conditions such as wind and rain. In this section,
we describe and define the metrics that were used to understand the variability of the detection algorithm
performance.
In general, more light results in better detection. The light condition can be varied by variation of a set
of factors such as weather conditions or amount of sediment which is carried by the river. In any case, the
daylight is a factor that can change the light condition systematically, *i.e.* low light early in the morning (Fig
12.a), bright light at midday with potential for direct light and shadows (Fig 12.b), and low light again in the
evening, though different from the morning because the hue is more bluish (Fig 12.c). This effect of the time
of day was quantified simply by noting the time of the image, which was marked on the top of each frame of



the recorded videos.

**Fig 12**

Detection is also strongly affected by the frame 'roughness', defined here as the variation in light over
small distances in the frame. The change in light is important for the recognition of wood objects, but light
roughness can also occur when there is a region with relatively light pixels due to something such as reflection
of the surface of the water, and dark roughness can occur when there is a region with relatively dark pixels
due to something such as shadows from the surface water waves. Detecting wood is typically more difficult
around light roughness, which results in false negatives, while the color-map of a darker surface is often close
to that of wood, which results in false positives. Both of these conditions can be seen in Fig 12 which is
highlighted in Fig 12.a.  In general, the frame roughness increases in windy days or when there is an obstacle
in the flow, such as downstream of the bridge pier in the current case. The light roughness was calculated for
the current study by defining a light intensity threshold and calculating the ratio of pixels of higher value
among the frame. The dark roughness is calculated in the same way, but in this case the pixels less than the
threshold were counted. In this work thresholds equal to 0.9 and 0.4 were used for light and dark roughness,
respectively.
The oblique view of the camera means that the distance of the wood piece from the camera is another
important factor in detection (Fig 13). The effect of distance on detection interacts with wood length, *i.e.*
shorter pieces of wood that are not detectable near the camera may not be detectable toward the far bank due
to the pixel size variation (Ghaffarian et al., 2020). Moreover, if a piece of wood passes through a region
with high roughness (Fig 13) or amongst bushes or trees (Fig 13 right hand side) it is more likely that the
software is unable to detect it. In our case, one day of video record could not be analyzed due to the presence
of a spider that moved around in front of the camera.

**Fig 13**

Flow discharge is another key variable in wood detection. Increasing flow discharge generally means
that water levels are higher, which brings wood close to the near bank of the river closer to the camera.  This
change can make small pieces of wood more visible, but it also reduces the angle between the camera position
and pixels, which makes wood farther from the camera harder to see.  High flows also tend to increase surface
waves and velocity, which can increase the roughness of the frame and lead to the wood being intermittently
submerged or obscured. More suspended sediment is carried during high flows which can change water



surface color and increase the opacity of the water.

## 4.2. Detection performance

Automatic detection software performance was evaluated based on the event based $TP$, $FP$, and $FN$
raw numbers and the precision (PR) and recall rates (RR) using the default parameters in the software. On
average, manual annotation resulted in the detection of approximately twice as many wood pieces as the
detection software (Table 3). Measured over all the events, RR = 29%, which indicates that many wood
objects were not detected by the software, while among detected objects about 36% were false detections
($PR = 64\%$).

**Table 3**

To better understand model performance, we first tested the correlation between the factors identified
in the previous section (Table 4). As shown, the pairs of dark/light roughness, length/distance and dis-
charge/time were highly correlated ($Corr. = 0.59, 0.46, 0.37$ respectively). For this reason, they were con-
sidered together to evaluate the performance of the algorithm within a given parameter space. The X/Y po-
sitions were also considered as a pair despite a relatively low correlation (0.15) because they represent the
position of an object. As a note, the correlation between time and dark roughness is higher than discharge
and time, but we used the discharge/time pair because discharge has a good correlation only with time.  As
recommended by MacVicar and Piégay (2012), wood lengths were determined on a log base 2 transformation
to better compare different classes of floating wood, similar to what is done for sediment sizes.

**Table 4**
**Fig 14**

The presentation of model performance by pairs of correlated parameters clarifies certain strengths and
weaknesses of the software (Figure 14). As expected, the results of Fig 14.b indicate that first, the software
is not so precise for small pieces of wood (less than the order of 1 m), and second there is an obvious link
between wood length and the distance from the camera so that by increasing the distance from the camera,
the software is precise only for larger pieces of wood. Based on Fig 14.e, the software precision is usually
better on the right side of the frame than the left side. It would be reasonable, as the software requires to
detect a patch at least in 5 continuous frames to recognize it as a piece of wood (see Sect 2.2 and Fig 4 for
more information). Therefore, most of the true positives are on the right side of the frame, where 5 continuous
frames have already established. Also, the presence of the bridge pier (at X ≅ -30 to -40 m based on Fig 10)





in the upstream, produces lots of waves that decreases the precision of the software. Also, Fig 14.h shows
that the software is much more precise during the morning when there is enough light rather than evening
when the sunshine decreases. However, at low flow ($Q < 550 \ m^3/s$) the software precision decreases sig-
nificantly. Finally, based on Fig 14.k, the software does not work well in two roughness conditions: (i) in a
dark smooth flow (light roughness $\cong 0$) when there are some dark patches (shadows ) on the surface (dark
roughness $\cong 0.3$), and (ii) when both roughness increases and there are many noises in a frame.
To estimate the fraction of wood pieces that the software did not detect, the recall rate $RR$ is calculated
in different conditions and a linear interpolation was applied on $RR$ as it is presented in Fig 14, third column.
According to Fig 14.c, $RR$ is fully dependent on piece length so that for the lengths at the order of 10 m ($L =$
$O(10)$) $RR$ is very good. By contrast when $L = O(0.1{\sim}1)$ the $RR$ is too small. There is a transient region
when $L = O(1)$ which is slightly depends on the distance from the camera. One can say, the wood length is
the most crucial parameter that affects the recall rate independent of the operator annotation. Based on Fig
14.f, the $RR$ is much better on the left side of the frame than on the right side. It can be because the operator's
eye needs some time to detect a piece of wood, so most of the annotations are on the right side of the frame.
Having a small number of detections on the left side of the frame results in the small value of $FN$ which
followed by high values of $RR$ in this region ($RR = TP/(TP + FN)$). Therefore, while the position of detec-
tion plays a significant role in the recall rate, it is completely dependent on the operator bias. By contrast,
frame roughness, daytime, and flow discharge do not play a significant role in the recall rate (Fig 14. i, l).

### 4.3. Post-processing

This section is separated into two main parts. First, we show how to improve the precision of the soft-
ware by a posteriori distinction between $TP$ and $FP$. After removing $FPs$ from the detected pieces, in the
second part, we test a process to predict the annotated data that the software missed *i.e.* false negatives.

### 4.3.1. Precision improvement

To improve the precision of the automatic wood detection we first ran the software to detect pieces and
extracted the eight key parameters for each piece as described in Sect 4.1.3. We then estimated the total
precision of each object, as the average of four precisions from each sub-figure of Fig 14. In the current study
the detected piece was considered to be a true positive if the total precision exceeded 50%. To check the
validity of this process, we used cross-validation by leaving one day out, calculating the precision matrices
based on five other days, and applying the calculated $PR$ matrices on the day that was left out. As is seen in



Table 5, this post-processing step increases the precision of the software to 85%, an enhancement of 21%.
The degree to which the precision is improved is dependent on the day left out for cross-validation. If, for
example, the day left out had similar conditions to the mean, the $PR$ matrices were well trained and were able
to distinguish between $TP$ and $FP$ (*e.g.* $2^{nd}$ Jan with 42% enhancement). On the other hand, if we have an
event with new characteristics (*e.g.* very dark and cloudy weather or at discharges different from what we
have in our database), the PR matrices were relatively blind and offered little improvement (*e.g.* $15^{th}$ Dec
with 10% enhancement).

**Table 5**

One difficulty with the post-processing reclassification of wood piece is that this new step can also

introduce error by classifying real objects as false positives (making them a false negative) or vice-versa.
Using the training data, we were able to quantify this error and categorized them as post-processed false
negatives ($\boldsymbol{FN_{pp}}$) with an associated recall rate ($\boldsymbol{RR_{pp}}$). As shown in Table 5, the precision enhancement
process lost only around 14% of $TPs$ ($RR_{pp}$= 86%).

Instead of using all eight key parameters (four $PR$ matrices) to calculate the overall precision, it is also

possible to use other configurations by combining different matrices as it is shown in Fig 15. In this figure,
the precision matrices 1 to 4 are the same as the matrices presented in Fig 14 and different colors show
different combinations of these matrices. As it is seen, some configurations (e.g. (2,4) or (1,3,4)) result in
better precision and some cases (e.g. (1,2) or (1,3)) there is almost no difference between post-processed $PR$
and the raw data. The reason that configurations like (2,4) or (1,3,4) with a better precision rate were not used
here was that in these cases the post-processed recall rate $RR_{pp}$ was low (around 60%) meaning that by using
these configurations many of true positives was removed. Therefore, to have the best precision enhancement
with maximum post-processed recall rate all 4 different precision matrices are used (Fig 15, dark red scatters).

**Fig 15**

**4.3.2.  Modeling missed wood pieces based on the recall rate**

The automated software detected 29% of the number of manually annotated wood pieces (Table 5). In

the previous section, it was described how to enhance the precision of the software to ensure that these auto-
matically detected pieces are $TPs$. The larger question, however, is how to model the missing pieces. Based
on Fig 14, the software work well for very large objects in most areas of the image and in most lighting
conditions. However, the smaller pieces were found to be harder to detect, making the wood length the most



important factor governing the recall rate. Based on this idea, the final step in the post processing is to apply
a model to account for the smaller wood pieces.
The model is based on the concept of a threshold piece length. Above the threshold, wood pieces are
likely to be accurately counting using the automatic software. Below the threshold, on the other hand, the
automatic detection software is likely to deviate from the manual counts. The actual length distribution was
first determined based on the manual annotations ($TP + FN$) (Fig 16.a). Also shown are the raw results of
the automatic detection software ($TP + FP$) and the raw results with the false positives removed ($TP$). At
this stage, the difference between the $TP$ and the $TP + FN$ lines are the false negatives ($FN$) that the software
has missed. Comparison between the two lines shows that they tend to deviate between 2-3 m. The correla-
tion coefficient between them was calculated for thresholds varying from 1 cm to 15 m length and 2.5 m
length was defined as the optimum threshold length for recall modeling (Fig 16.b).
In the next step we wanted to estimate the pieces less than 2.5 m that the software missed. During the
automatic detection process, when the software detects a piece of wood, according to Fig 14 (third column),
the $RR$ can be calculated for this piece (same protocol as precision enhancement in Sect 4.3.1). Therefore, if
for example the average recall rate for a piece of wood is 50%, there is likely to be another piece in the same
condition (defined by the eight different parameters described in Table 2) that the software could not detect.
To correct for these missed pieces, additional virtual pieces were added to the database. Fig 16.a, shows the
length distribution after adding these virtual pieces to the database (blue line, total of 5841 pieces). The result
shows a good agreement between this and the operator annotations (green line, total of 6249 pieces), which
results in a relative error of only 6.5% in the total number of wood pieces.

**Fig 16**

On the Ain River by separating videos to 15 min segments, MacVicar and Piégay, (2012) and Zhang et
al., (2020) proposed the following equation for calculating wood discharge from the wood flux:
$$Q_w = 0.0086F^{1.24} \tag{1}$$
where, $Q_w$ is the wood discharge ($m^3/15min$) and $F$ is the wood flux (piece number/15 min). Using
this equation, the total volume of wood was calculated based on three different conditions: (i) operator anno-
tation ($TP + FN$), (ii) raw data of the detection software ($TP + FP$) and (iii) post-processed data of the de-
tection software ($TP_{modeled}$). Fig 17 shows a comparison of the total volume of wood from the manual



annotations in comparison with the raw and post-processed annotations from the detection software. As
shown, the raw detection results underestimate wood volume by almost one order of magnitude. After pro-
cessing, the results show some scatter but are distributed around the 1:1 slope, which indicates that they
follow the manual annotation results. There is a slight difference for days with lower fluxes (Jan 4 and 7),
where the post-processing tends to over-estimate wood volumes, but in terms of an overall wood balance the
volume of wood on these days are negligible. In total, 125 $m^3$ wood was annotated by the operator and the
software automatically detected only 46 $m^3$, some of which represent false positives. After post-processing,
142 $m^3$ wood was estimated to have passed in the analyzed videos for a total error of 13.5%.

**Fig 17**

**5.    Conclusion**
Here, we present new software for the automatic detection of wood pieces on the river surface. After
presenting the corresponding algorithm and the user interface, an example of automatic detection was pre-
sented. We annotated 6 days of flood events that were used to first check the performance of the software
and then develop post-processing steps to both remove possibly erroneous data and model data that were
possibly missed by the software.
To evaluate the performance of the software, we used precision and recall rates. The automatic detection
software detects around one third of all annotated wood pieces with 64% precision rate. Then using the op-
erator annotations as the ultimate goal, the post-processing part was applied to extrapolate data extracted
from detection results, aiming to come as close as possible to the annotations. It is shown that using four pair
of key factors: (i) light and dark roughness of the frame, (ii) daytime and flow discharge, (iii) X, Y coordinates
of detection position, and (iv) distance of detection as a function of piece length, it is possible to detect false
positives and increase the software precision to 86%. Using the concept of a threshold piece length for de-
tection it is shown that it is then possible to model the missed wood pieces (false negatives). In the presented
results, the final recall rate results in a relative error of only 6.5% for piece number and 13.5% for wood
volume.
This work shows the feasibility of the detection software to detect wood pieces automatically. Auto-
mation will significantly reduce the time and expertise required for manual annotation, making video moni-
toring a powerful tool for researchers and river managers to quantify the amount of wood in rivers. The

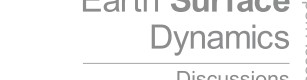

software should be applied in other rivers to test it in different contexts and enhance its accuracy.

## 6. Code/Data/Sample availability

Not available.

## 7. Author contribution

Hossein Ghaffarian: Application of statistical, and computational techniques to analyses study data. Creation
and presentation of the published work.
Hervé Piégay, Bruce MacVicar, Hossein Ghaffarian: Development and design of methodology; creation of
models.
Laure Tougne, Pierre Lemaire: Programming and software development.
Pierre Lemaire, Zhang Zhi: Performing the surveys, and data collection.
Hervé Piégay, Bruce MacVicar, Pierre Lemaire, Hossein Ghaffarian: Critical review, commentary, and revi-
sion.
Hervé Piégay: Oversight and leadership responsibility for the research activity planning and execution, in-
cluding mentorship external to the core team.

## 8. Competing interests

The authors declare that they have no conflict of interest.

## 9. Acknowledgment

This work was performed within the framework and with the support of the PEPS (RiskBof Project (
2016)) and LABEX IMU (ANR-10-LABX-0088) and within the framework of the EUR H2O'Lyon (ANR-
17-EURE-0018) of Université de Lyon, le latter being both part of the program "Investissements d'Avenir"
(ANR-11-IDEX-0007) operated by the French National Research Agency (ANR).





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

ing of in-channel wood fluxes: critical events, flux prediciton and sampling window. *Earth Surface Processes*
*and Landforms*





**Table 1 Characteristics of streamside video monitoring techniques in different studies.**

| Article | Sampling | Temporal scales | Camera resolution | Study site |
|---|---|---|---|---|
| MacVicar & Piégay (2012) | 15 min segments | 3 floods/18 hr/5 fps | 640 × 480 | Ain, France |
| Kramer & Wohl (2014) | Total duration | 32 days/12761 frames/0.017 fps | n/a | Slave, Canada |
| Boivin et al. (2017) | Total duration | 3 floods/150 hr/25 fps | 640 × 480 | St Jean, Canada |
| Kramer et al. (2017) | Total duration | 11 months/0.0017 fps | 1268 × 760 | Slave, Canada |
| Senter et al. (2017) | 15 min segments | 39 days/180 hr/4 fps | 2048 × 1536 | North Yuba, USA |
| Ghaffarian et al. (2020) | Total duration | 2 floods/80 hr/1 fps | 600 × 800 | Isère, France |
| Zhang et al. (2020) | Total duration | 7 floods & 1 windy period /183 hr/5 fps | from 640 × 480 up to 800 × 600 | Ain, France |

**Table 2 Parameters used to assess the performance of the software**

| Parameter | Rational | Metric |
|---|---|---|
| Piece length | Larger objects are easier to detect. | Detecting an object in pixel coordinates. Transferring coordinates to metric. Calculating length, position, and distance. |
| Distance | Objects closer to the camera are easier to detect. | |
| X position | Some particular areas of turbulent flow in the field of view affect detection (e.g. presence of a bridge pier). | |
| Y position | | |
| Discharge | Flow discharge affects water color, turbulence and the amount of wood. | Recorded water elevation data and calibrated rating curve at hydrologic station. |
| Time | Luminosity of the frames varies with time of day. | Time of day as indicated on top of each frame. |
| Dark roughness | Small spots with sharp contrast (either lighter or darker) affect detection. | % of pixels below an intensity threshold |
| Light roughness | | % of pixels above an intensity threshold |


**Table 3 Summary of automated and manual detections**

| Date | discharge ($m^3/s$) | | Water level ($m$) | | Detection time ($hr$) | Number | | Precision rate% | Recall rate% |
|---|---|---|---|---|---|---|---|---|---|
| | $Q_{max}$ | $Q_{min}$ | $h_{max}$ | $h_{min}$ | | annot. | det. | | |
| 1/1/2012 | 718 | 633 | -7.4 | -7.8 | 7 to 17 | 2282 | 972 | 77 | 33 |
| 2/1/2012 | 772 | 674 | -7.2 | -7.6 | 7 to 17 | 802 | 380 | 52 | 24 |
| 4/1/2012 | 475 | 423 | -8.4 | -8.6 | 7 to 17 | 140 | 158 | 20 | 22 |
| 6/1/2012 | 786 | 763 | -7.2 | -7.2 | 7 to 17 | 712 | 384 | 54 | 29 |
| 7/1/2012 | 462 | 430 | -8.5 | -8.6 | 7 to 17 | 117 | 73 | 40 | 25 |
| 15/12/2012 | 707 | 533 | -7.5 | -8.2 | 9 to 14 | 1296 | 503 | 72 | 28 |
| Total | 786 | 423 | -7.2 | -8.6 | 55 hr | 5349 | 2470 | 64 | 29 |






**Table 4 Correlation between parameters**

|  | Dark roughness | Light roughness | Length | Distance | X position | Y position | Discharge | Time |
|---|---|---|---|---|---|---|---|---|
| Dark roughness |  | 0.59 | -0.02 | -0.04 | 0.04 | 0.1 | 0 | 0.57 |
| Light roughness | 0.59 |  | -0.03 | -0.03 | 0.03 | 0.09 | -0.04 | 0.29 |
| Length | -0.02 | -0.03 |  | 0.46 | -0.45 | -0.35 | -0.02 | -0.01 |
| Distance | -0.04 | -0.03 | 0.46 |  | -1 | -0.16 | 0.14 | -0.05 |
| X position | 0.04 | 0.03 | -0.45 | -1 |  | 0.15 | -0.15 | 0.05 |
| Y position | 0.1 | 0.09 | -0.35 | -0.16 | 0.15 |  | 0 | 0.07 |
| Discharge | 0 | -0.04 | -0.02 | 0.14 | -0.15 | 0 |  | 0.37 |
| Time | 0.57 | 0.29 | -0.01 | -0.05 | 0.05 | 0.07 | 0.37 |  |

**Table 5 Precision rate (PR) before and after post-processing**

|  |  | 1 Jan | 2 Jan | 4 Jan | 6 Jan | 7 Jan | 15 Dec | Total |
|---|---|---|---|---|---|---|---|---|
| Raw data | $TP$ | 745 | 196 | 31 | 206 | 29 | 363 | 1570 |
|  | $FP$ | 227 | 184 | 127 | 178 | 44 | 140 | 900 |
|  | $FN$ | 1537 | 606 | 109 | 506 | 88 | 933 | 3779 |
|  | $PR\%$ | 77 | 52 | 20 | 54 | 40 | 72 | 64 |
|  | $RR\%$ | 33 | 24 | 22 | 29 | 25 | 28 | 29 |
| Post-proc. | $TP$ | 658 | 150 | 30 | 178 | 22 | 315 | 1353 |
|  | $FP$ | 64 | 10 | 60 | 39 | 11 | 68 | 252 |
|  | $FN_{pp}$[1] | 87 | 46 | 1 | 28 | 7 | 48 | 217 |
|  | $PR\%$ | 91 | 94 | 33 | 82 | 67 | 82 | 85 |
|  | $RR_{pp}$[2]$\%$ | 88 | 77 | 97 | 86 | 76 | 87 | 86 |
| $PR$ improvement |  | 14 | 42 | 13 | 28 | 27 | 10 | 21 |

[1] $FN_{pp}$ denotes the false estimations of the precision matrices which results in missing some $TP$.
[2] $RR_{pp}$ denotes the recall rate of post processing which corresponds to $FN_{pp}$.




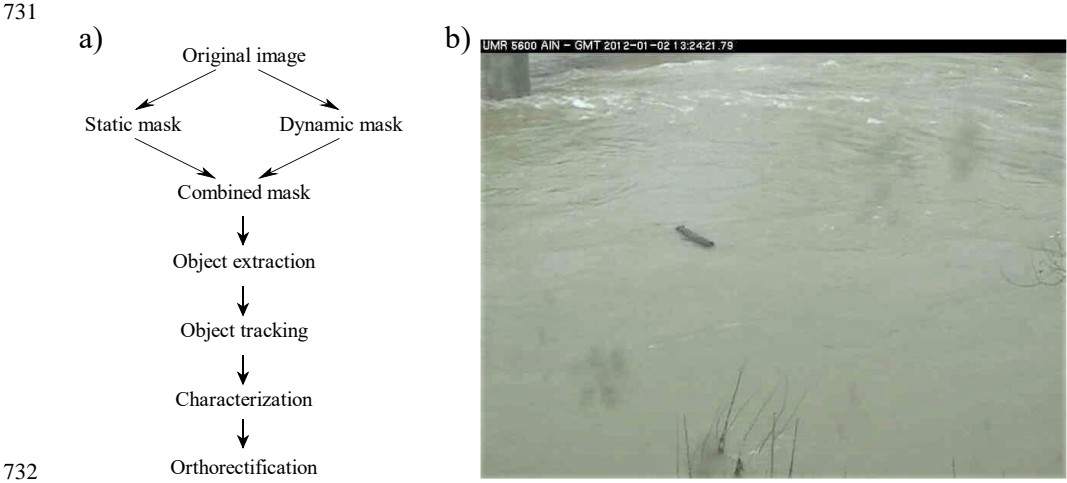

**Fig 1 a) Flowchart of the detection software and b) an example of frame on which these different flowchart steps are applied.**


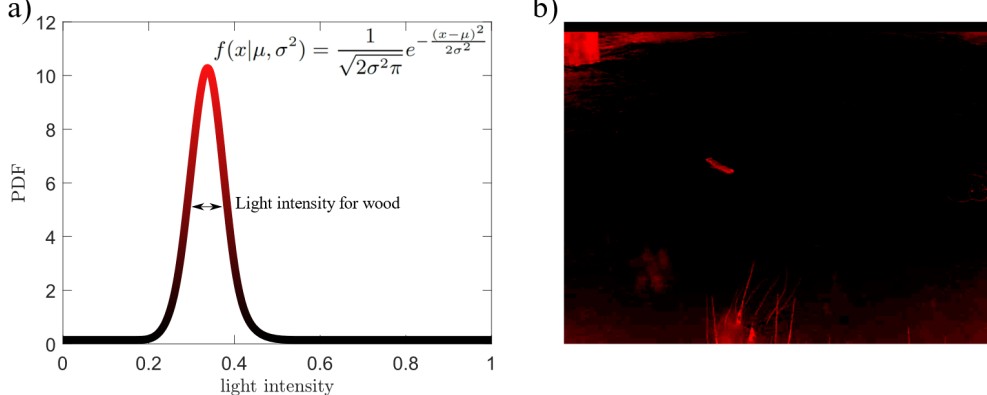


**Fig 2 Static probability mask, a) Gaussian distribution of light intensity range for a piece of wood, b) employment of probability mask on the sample frame.**



a)

b)

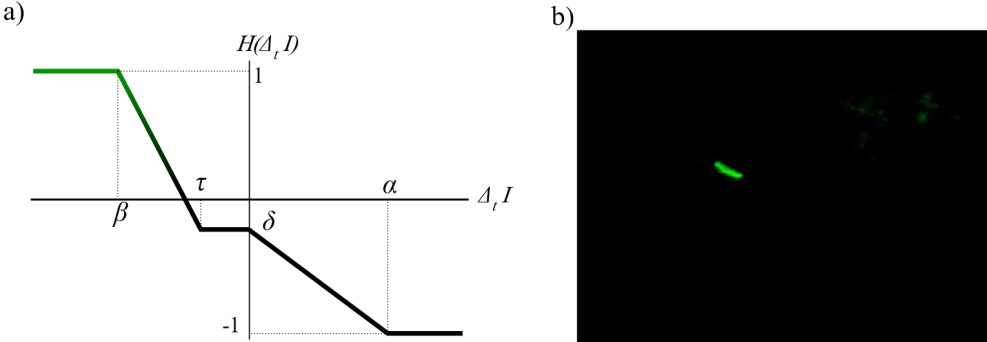


**Fig 3 Dynamic probability mask, a) updating function $H(\Delta_t, I)$ adapted from Ali et al. (2011) and b) employment of probability mask on the sample frame.**

a)

b)

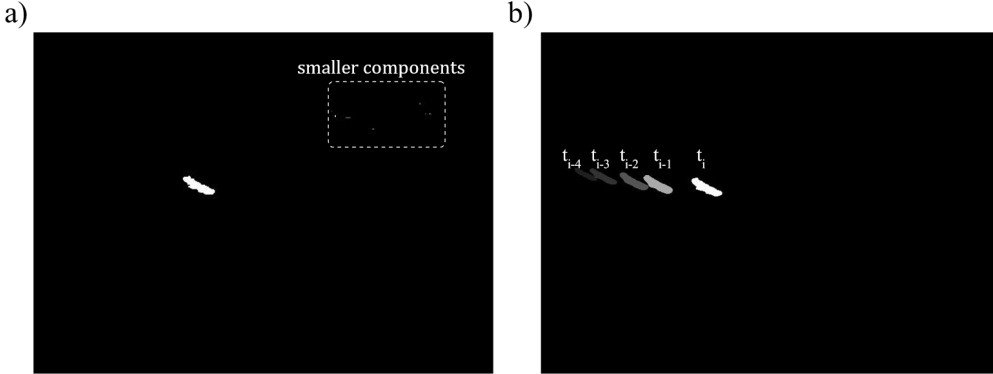


**Fig 4 a) Object extraction by (i) combining static and dynamic masks and (ii) applying a threshold to retain only high-probability pixels. b) Object tracking as a filter to deal with partly immersed objects and to distinguish between moving objects from static waves.**



Earth **Surface**
Dynamics
Discussions




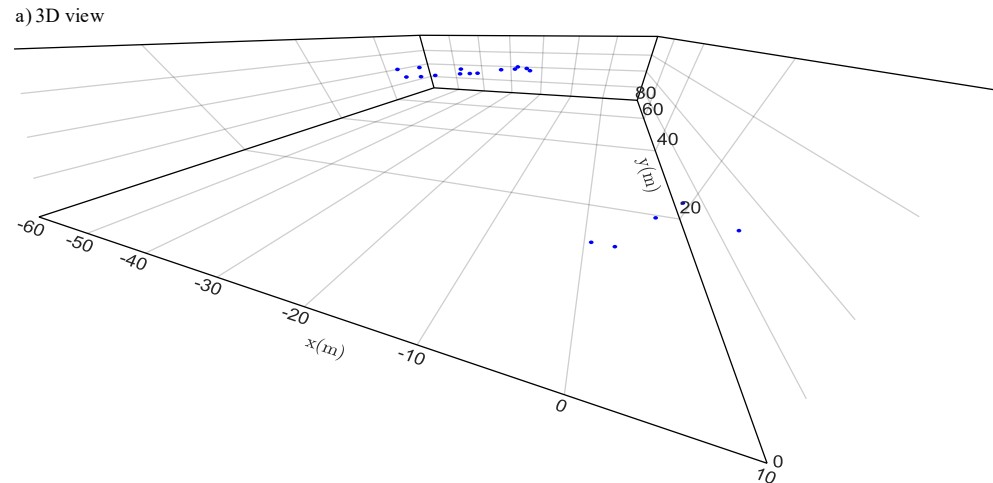

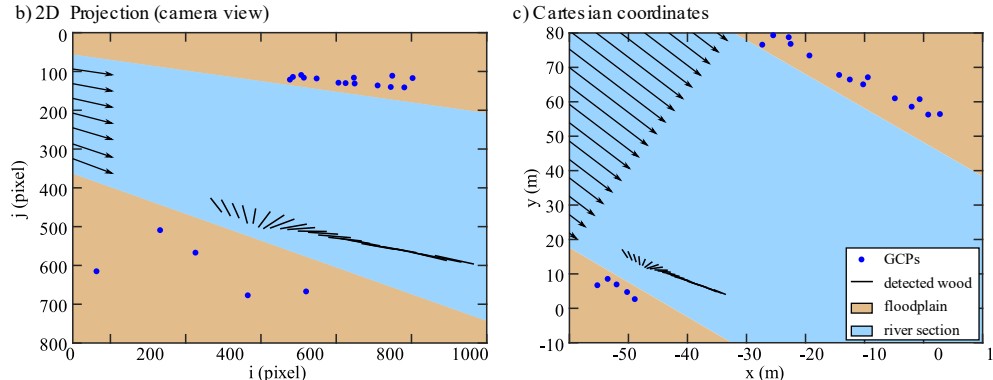


**Fig 5 Image rectification, process. 3D view of non-colinear GCPs in metric coordinates (a), their corresponding localization within the image (b), and the relative 2D metric coordinates for a given water level (c). (b,c) A practical example of the transformation of the coordinates is presented. The different solid lines represent the successive detection in a set of consecutive frames.**




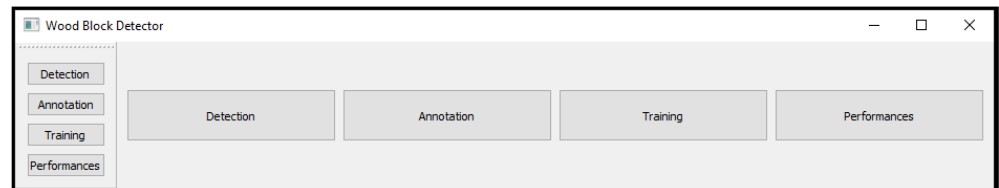

**Fig 6 User interface of the detection software.**

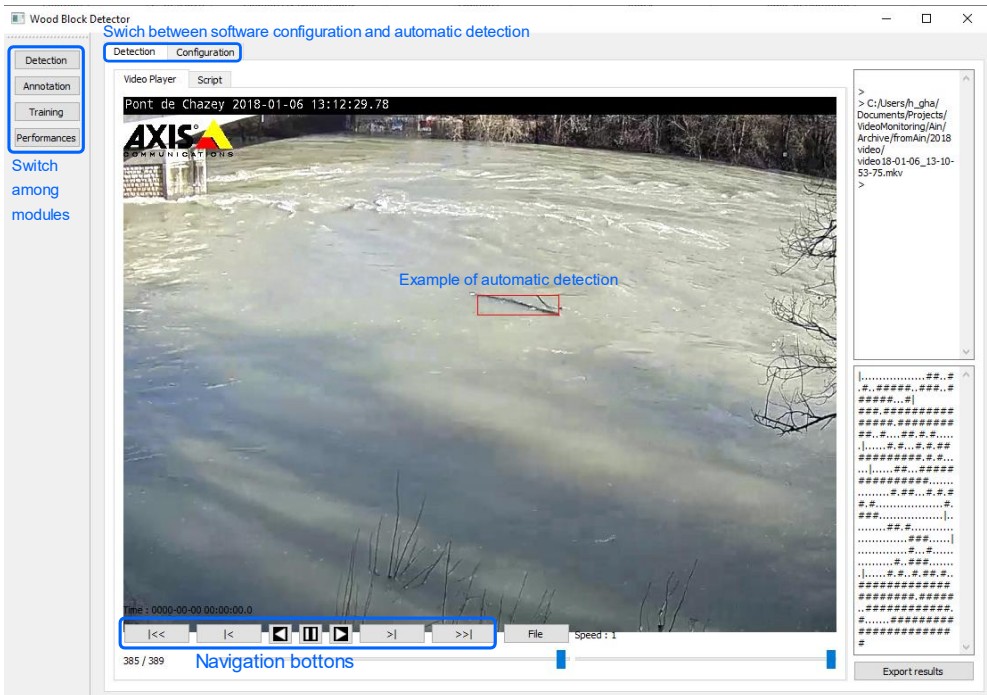


**Fig 7 User interface of the detection module of automatic detection software.**



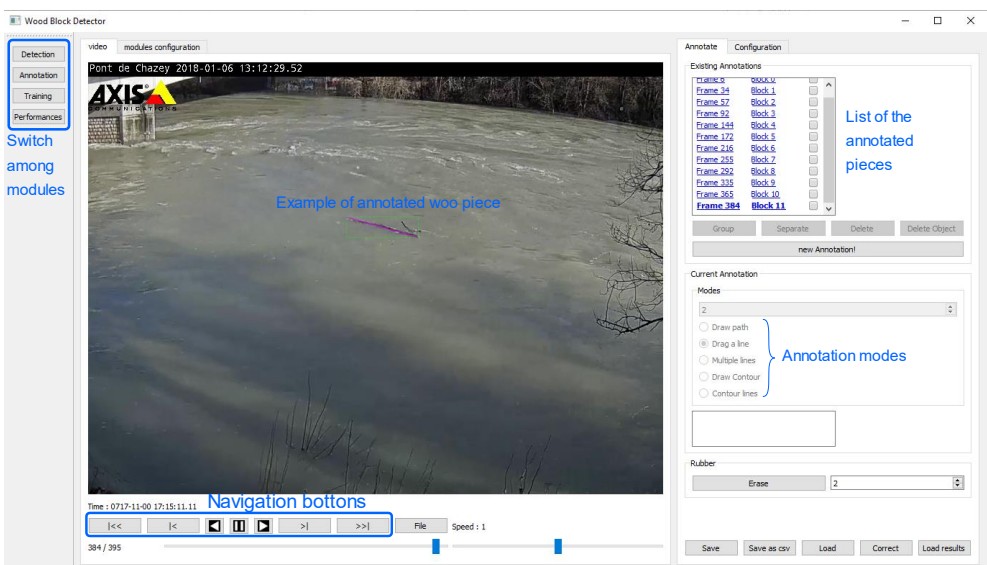

**Fig 8 User interface of the annotation module of automatic detection software.**


Earth **Surface**
**Dynamics**
Discussions

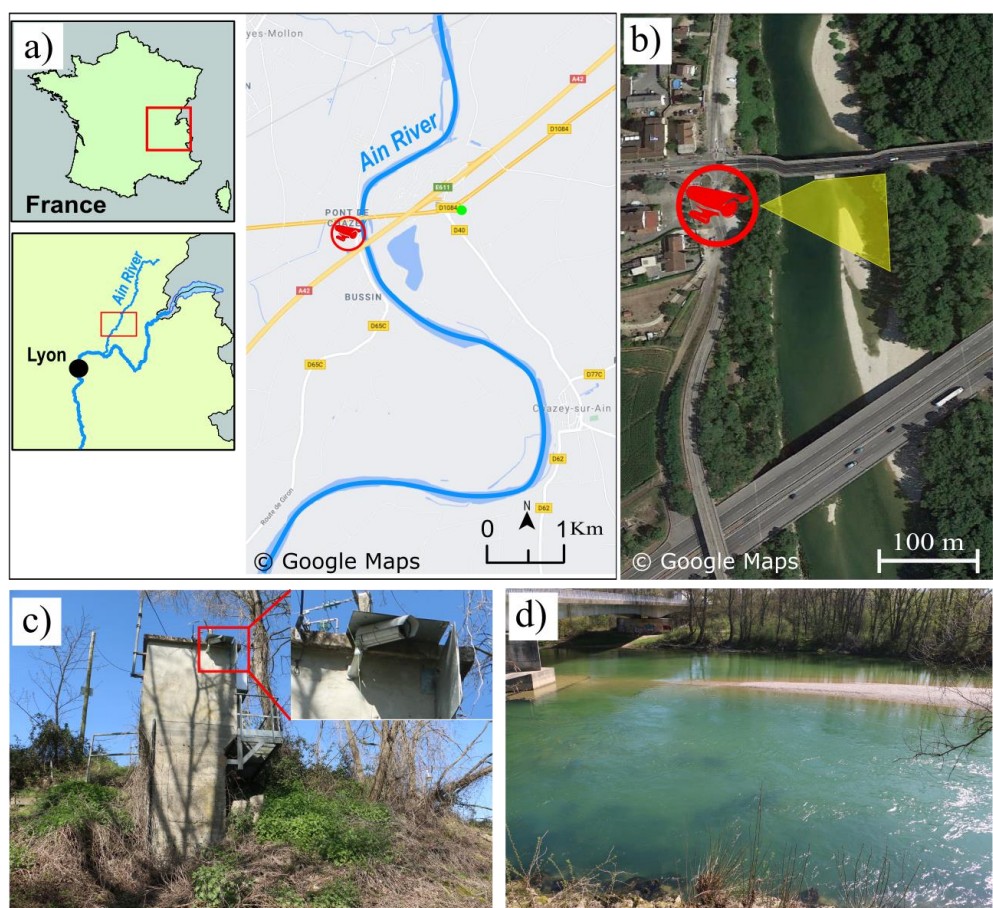


**Fig 9 Study site at Pont de Chazey: a) Location of the Ain River catchment in France and location of the gauging and meteorological stations, b) camera position and its view angle in yellow, c) overview of the gauging station with the camera installation point.**



Earth **Surface**
**Dynamics**
Discussions



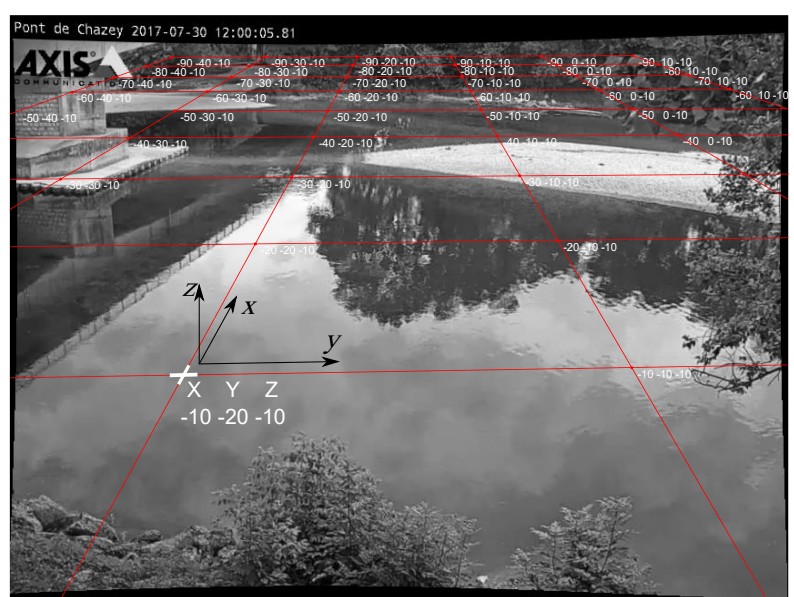


**Fig 10 Rectifying transformation matrix at low flow level with camera at (0,0,0).**

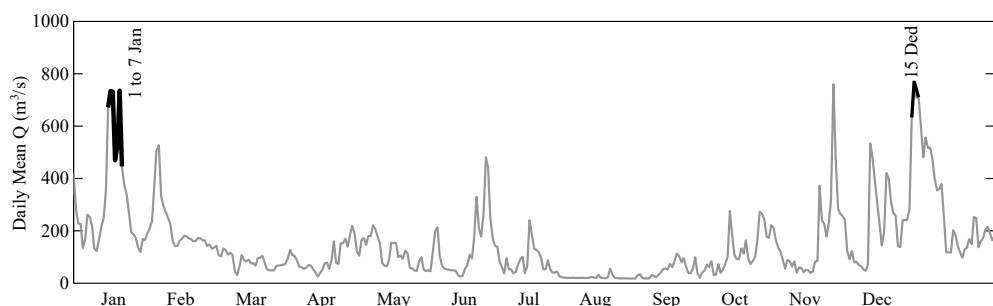


**Fig 11 Daily mean discharge series for monitoring period from 1st to 7th January and in 15th December.**



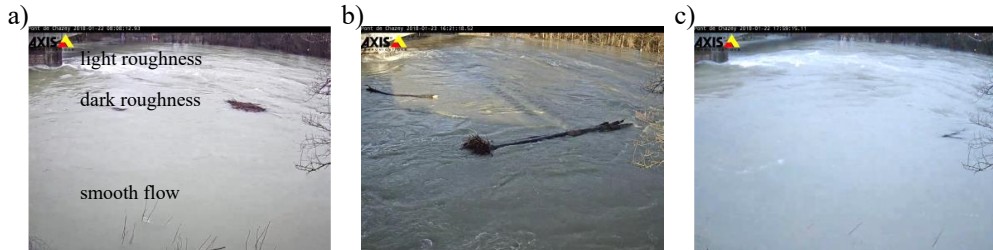


**Fig 12 Different light conditions during (a) morning, (b) noon and (c) late afternoon, results in different frame roughness's and different detection performances.**

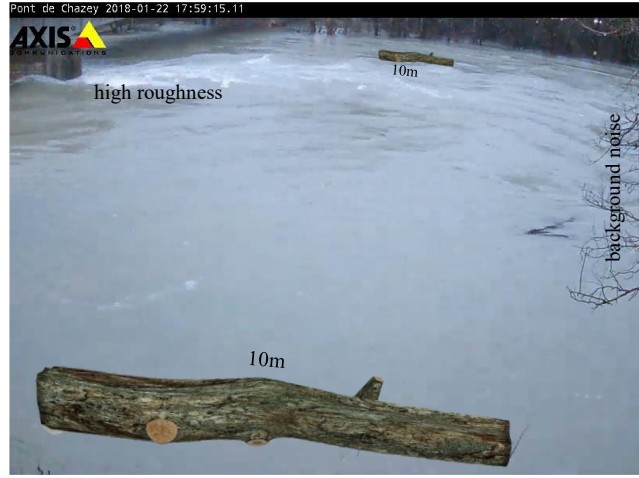


**Fig 13 Wood position can highly affect the quality of detection. Pieces that are passing in front of the camera are detected much better than the pieces far from the camera.**

Earth **Surface**
**Dynamics**
Discussions

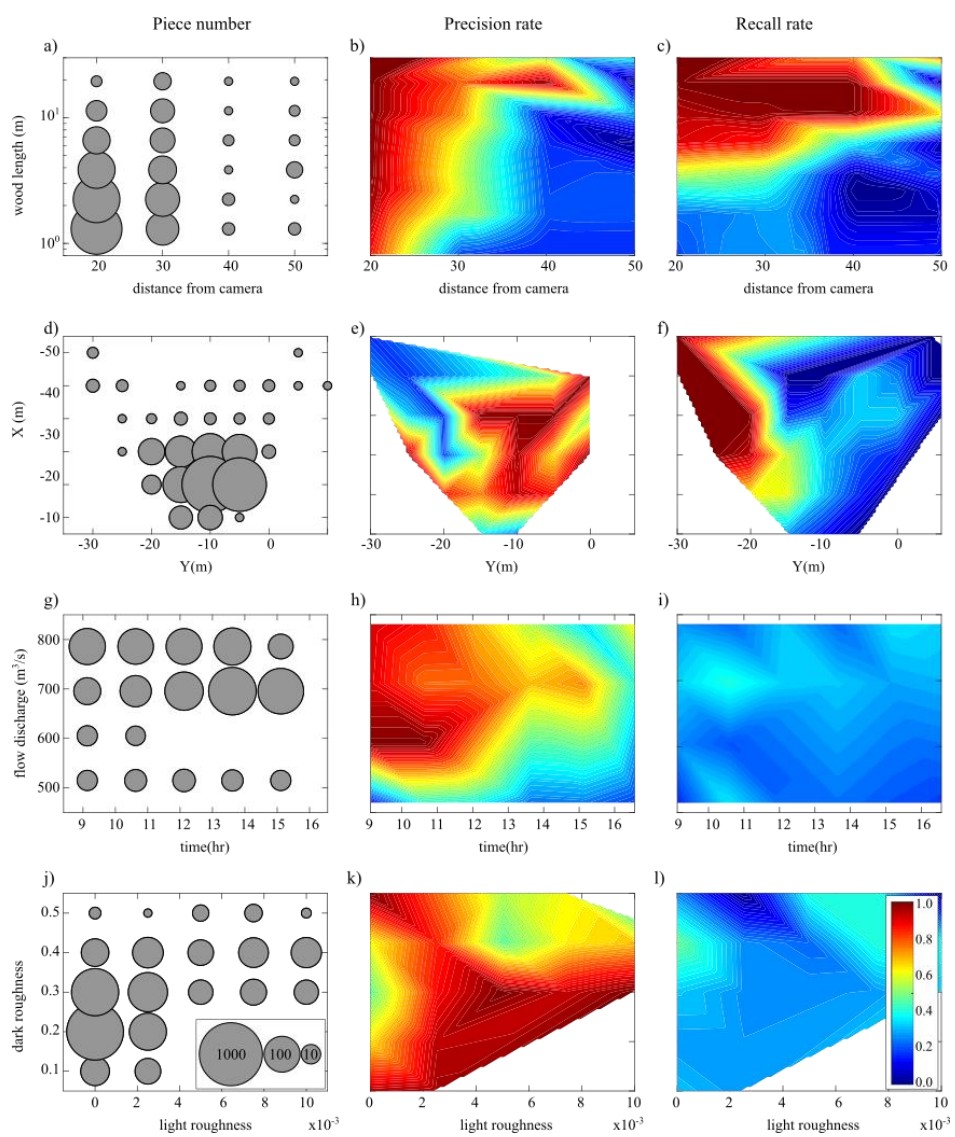


**Fig 14 Correction matrices: a, b, c) wood lengths as a function of the distance from the camera, d, e, f) detection position, g, h, i) flow discharges during the daytime, and j, k, l) light and dark roughness's. The first column shows number of all annotated pieces. Second and third columns show Precision and Recall rates of the software respectively.**



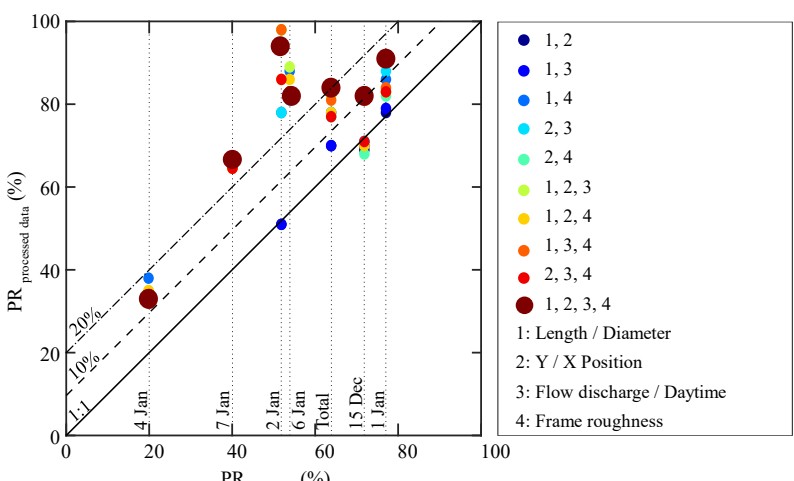

**Fig 15 Effect of using different combinations of *PR* matrices on precision improvement compared with 1:1 line(no improvement), 10% and 20% improvement lines.**

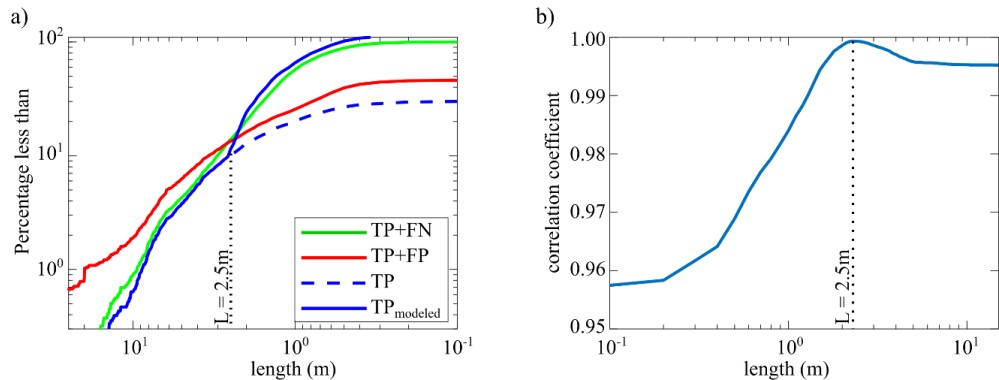

**Fig 16 a) Steps to post-process software automatic detections: (i) raw detections (*TP* + *FP* red line), (ii) Only true positives using the *PR* improvement process (*TP* blue dashed line), and (iii) modeling false negatives (blue line). Operator annotation (green line is used as a benchmark). b) The correlation coefficient between operator annotation and modeled *TP* to find an optimum threshold length for *RR* improvement.**



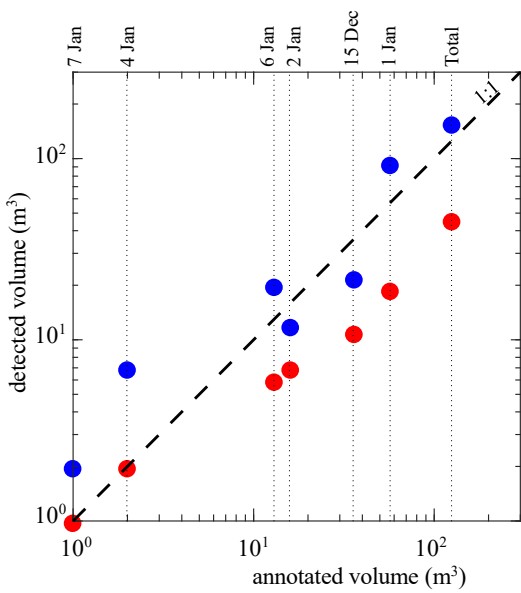


**Fig 17 Comparison of the total volume of wood between operator annotation as the benchmark and raw data (red scatters) and post-processed data (blue scatters), compared with a 1:1 line.**