# Peer review of "Automated quantification of floating wood pieces in rivers from video monitoring: a new software tool and validation."

_Earth Surface Dynamics, 2020_

## Referee Comment (RC1) · Anonymous Referee #1 · 11 Jan 2021

General comments: This manuscript presents a framework to automatically detect floating large wood in video records. A software has been developed and its performance has been evaluated comparing the automatic detection with manual identification, which is very time consuming. Applying this promising method would make wood transport monitoring much easier in any river. Therefore, the work is of high interest and provides an important contribution to river science and within the scope of Earth Surface Dynamics. I have several comments and suggestions, though. Firstly, the general structure of the manuscript is a bit confusing, with several sections about methodological aspects (i.e., section 2, 4.1. . .) and several of them mix methods with results and discussion. I would recommend structuring the manuscript in a better way,

by splitting methods, from results from discussion. Secondly, some details are needed for some unclear parts, some figures need to be edited (or even removed) and some misunderstanding about modelling need to be clarified. Specific comments: In the Abstract, authors claim that their approach may work potentially in real time, this is a very promising challenge and the post process which seems to be required makes the real time analysis a bit less realistic. . .so, maybe this is something to be discussed in the discussion, but smoothed in the abstract. What do the authors mean by human-month? Line 43: you may replace the cited conference paper by Ruiz-Villanueva et al., 2018, by the journal paper: Ruiz-Villanueva, V., Mazzorana, B., Bladé, E., Bürkli, L., Iribarren-Anacona, P., Mao, L., Nakamura, F., Ravazzolo, D., Rickenmann, D., Sanz-Ramos, M., Stoffel, M., Wohl, E., 2019. Characterization of wood-laden flows in rivers. Earth Surf. Process. Landforms. doi.org/10.1002/esp.4603 Related to that, I was also wondering about the wood transport regime, which may play a role. In the Ain river, wood is transported as uncongested flow, sometimes maybe some cluster appear, as in semi-congested transport, but how the software would work with congested wood fluxes? Line 43 and several places throughout the manuscript: I could not find the reference: Zhang Z, Ghaffarian H, MacVicar B, Vaudor L, Antonio A, Michel K, Piégay H. 2020. Video monitoring of in-channel wood fluxes: critical events, flux prediction and sampling window. Earth Surface Processes and Landforms Is this a recently accepted/published paper, or is it under review? Or is it this one?: Zhi Zhang Z, Ghaffarian H, MacVicar B, Vaudor L, Antonio A, Michel K, Piégay H. 2020. Video monitoring of in-channel wood: from flux characterization and prediction to recommendations to equip stations. Earth Surface Processes and Landforms Please, correct it. Lines 91-107: this discussion about the use of computer science in general to detect wood is really interesting, and a question I was asking myself while reading, but it would go better in the discussion section. The same applies to the challenges of applying machine learning or deep learning approaches to images from rivers (lines 103-107), so, authors chose a different approach; but I think that the incredible and unique database from the Ain river, could indeed be used to develop Machine learning and deep learning algorithms that

can be applied to wood monitoring, right?. So, it could be useful to discuss about that in the discussion. The same about lines 201-204 and 208-209 (this statement about the good performance of the approach should better go to the discussion after the results are shown and can support that). The final step related to the wood characterization in terms of size and location is not very clear to me (Lines 210-218). Is the step manual? How the software can select the better view (e.g., when wood is less submerged)? Section 2.3 should start by explaining the image rectification, why this is necessary and how it was done, rather than discussing the loss of quality by warping (lines 220-222). Line 222: explain what the particular characteristics are of the images topology. Line 223: what other characteristics? Line 226: add a reference to the toolbox Line 241: a reference is needed here Line 259: the use of terms like "is intended to work" or "the idea is to load a video" sound like the software is currently not doing that. . .if so, I think it should be better to describe what the software actually does and discuss in the discussion section the prospects. I would structure Section 3 in a different way, following the order in which the work should be done. By describing in the first place the annotation, then learning and finally the automatic detection. Lines 278-285: are the different annotation methods all implemented in the software? What is the difference between them? How the use of one influence the results? Line 287: what do the authors mean by "as fine annotation as they wish"? I think the comparison of the annotation methods with other approaches should go in the discussion. In a previous line (line 249) four modules are mentioned (detection, annotation, learning and performance), but nothing is explained about learning later. Or is this training? Line 359: this is the first time that wood discharge is mentioned, please provide the units in which is measured 8only provided in Line 524). It could be useful to define also here wood flux. Table 3: total is the mean? Table 4: are all the correlations significant? If not, authors could use bold font to highlight the statistically significant ones. Lines 488-496: the use of post process matrices is not very clear to me, and neither the graph shown in Figure 15. In addition, in Figure 15 the x-axe label is cut. Section 4.3.2. modelling here is unclear, I think authors mean estimating or predicting. . . Line 501: the software works

or worked well. . .but, what do the authors mean by well? Line 504: model? What type of model? Although explained later, it sounds just like a threshold that is set up based on the manual annotation. This is related to my previous comment about modelling. I think better using estimation. Lines 508-509: not clear The conclusion section is just a summary of the previous content, while it should provide an outlook and state the most important outcome of your work. Do not simply summarize the work, but instead, interpret your findings at a higher level of abstraction. Further specific comments on Figures: Figure 5: is (a) really needed? Not sure about the additional information it conveys. Figure 6 to 8; are they really necessary? Figure 6 probably not. . .and 7 and 8 could be merged into one. The number of figures in the manuscript is really large. In order to reduce the number of figures I would suggest the following: • Figure 11 could be merged to Figure 9 (study site and studied period). • Figure 12 and 13 could also be merged. Figure 14: why e and f plots have different area? The same question about k and l. Figure 16: using only colours (especially red and green) could be an issue for colour blind readers, consider using also different line types. Figure 17: use different shapes for the two dot groups.

---

## Referee Comment (RC2) · Anonymous Referee #2 · 3 Feb 2021

General comments: In this work, the authors present a software that can automatically detect floating wood from video files of a river cross section. They introduce the processing steps of the software involving the formation of wood masks from training data followed by automatic detection and post-processing based on the recall rates of wood pieces. They discuss the factors that affect wood detection and present a quantitative analysis for automatically extracting wood from imagery. The automatic delineation of wood in imagery is not trivial and I believe this work would be of significant interest to readers of Earth Surface Dynamics and wood researchers. However, there are aspects of the manuscript that could be improved. Overall, I believe the work would benefit from further clarification of the masking procedure and the post-processing

steps. Given how important these steps are for accurate analysis, a more detailed description of these components would assist readers when using the software. For example, L151-153,283-285, how are images chosen for the annotation and how many are recommended? A recommendation or assessment of the amount of training data would assist future users of the software. In the post-processing (L475) it was unclear to me how the precision matrices were applied to the day of interest. Are these matrices generated for each piece of wood detected? What about the assessment of wood length between the manual annotations with the software? Also, at the times in the manuscript, it was unclear when you mention "it" or "object" what you were actually referring to- the software, a wood piece, the entire video set? I highlight some specific examples of this below. Specific comments: I may have missed it, but a link to the software or the details of how to access the software would be useful. L164: I was confused on your use of "stable" here. Wouldn't "stable" wood detections be false positives as they would actually be vegetation or the bridge being detected? L150 and 168: You first define variable x as the pixel light intensity and then again as position in the image. Please revise. L429: How did you measure the correlation between these parameters? L443: What is "it"? Technical comments: L209: Replace"by" with "be" L222: should be "calculate" L241: Please revise sentence L278: Should be "manual" L288-290: Some grammar issues, please revise. L452: Unclear what you mean by "many noises in a frame". Table 5: Is PR improvement also a percentage? L506: Replace "counting" with "counted" Fig 15: Horizontal axis label is cut off

---

## Author Comment (AC1) · 12 Mar 2021

The authors would like to appreciate the anonymous referee #1 for his/her general view point on their work. We wanted to point out that this paper is a methodological paper so that we could not follow the classical outline of a thematic paper (Material and methods, Results, Discussion). The method part is very detailed because it is the main contribution of such paper (the "result") and discussion is classically replaced here by a validation part / performance assessment. We read the methodological section and do not detect discussion or result points. There are a few references but they do not discuss the points but refer to methodological information we used in our protocol. The section is descriptive of what we did. The next section just introduced interface and again is a description of the tools which has been developed. There are also references here, not for discussions but as examples to refer to existing experiences to make explanations less theoretical. The next section is focused on validation to demonstrate our methodological approach is accurate. Perhaps the Material & Method part located in this last section may be the confusing point so that we remove it and create a short new section 2 introducing the study case (river and camera characters) and some parts dealing with rectifications have been reintroduced in the methodological section.

   In the following we tried to answer the comments one by one and we hope that the referee will find our answers acceptable. All the changes in the main manuscript concerning the replies to the referee #1 were highlighted in green.

**Specific comments:**

In the Abstract, authors claim that their approach may work potentially in real time, this is a very promising challenge and the post process which seems to be required makes the real time analysis a bit less realistic...so, maybe this is something to be discussed in the discussion, but smoothed in the abstract.

  The power of our postprocessing approach is that after we extract the correction matrices, which is a time-consuming process, the calculation time for PR enhancement is negligible. There are eight, easy access, parameters that based on them the code cab judge if a detection is TP or FP. To give an idea about the calculation time, here the time between each frame was 0.2s while the computer calculates total PR for one piece in 0.001s and for more than 6000 pieces of wood in 0.025s. Therefore,

though for the moment the detection code has not been ran in real time, and also the PR enhancement feature is not mounted on the software, soon it can be running in real time. We clarified this point in the text in conclusion section, so that the reader can understand real time is a realist step.

"Applying the post-process steps in real time is a realistic next step, because after we extract the correction matrices, which is a time-consuming process, the calculation time for PR, RR enhancement is negligible (less than 0.001s/piece)."

What do the authors mean by human-month?

It is changed to "over 150 labor-hours".

Line 43: you may replace the cited conference paper by Ruiz-Villanueva et al., 2018, by the journal paper: Ruiz-Villanueva, V., Mazzorana, B., Bladé, E., Bürkli, L., IribarrenAnacona, P., Mao, L., Nakamura, F., Ravazzolo, D., Rickenmann, D., Sanz-Ramos, M., Stoffel, M., Wohl, E., 2019. Characterization of wood-laden flows in rivers. Earth Surf. Process. Landforms. doi.org/10.1002/esp.4603

It is changed.

Related to that, I was also wondering about the wood transport regime, which may play a role. In the Ain river, wood is transported as uncongested flow, sometimes maybe some cluster appear, as in semi congested transport, but how the software would work with congested wood fluxes?

This software works based on two major characters of a floating piece of wood. Color contrast (static mask) and motion (dynamic mask) so if a patch on the surface has both these characters it can be detected as a piece of wood. It can be a boat, garbage, a cluster or one piece. This point is added to the conclusion as follow:

"It should be noted that the software cannot distinguish between a single piece of wood or the pieces in a cluster of wood in the congested wood fluxes."

Line 43 and several places throughout the manuscript: I could not find the reference: Zhang Z, Ghaffarian H, MacVicar B, Vaudor L, Antonio A, Michel K, Piégay H. 2020. Video monitoring of in-channel wood fluxes: critical events, flux prediction and sampling window. Earth Surface Processes and Landforms Is this a recently accepted/published paper, or is it under review? Or is it this one? Zhi Zhang Z, Ghaffarian H, MacVicar B, Vaudor L, Antonio A, Michel K, Piégay H. 2020. Video monitoring of in-channel wood: from flux characterization and prediction to recommendations to equip stations. Earth Surface Processes and Landforms Please, correct it.

It is corrected.

Lines 91-107: this discussion about the use of computer science in general to detect wood is really interesting, and a question I was asking myself while reading, but it would go better in the discussion section. The same applies to the challenges of applying machine learning or deep learning approaches to images from rivers (lines 103-107), so, authors chose a different approach; but I think that the incredible and unique database from the Ain river, could indeed be used to develop Machine learning and deep learning algorithms that can be applied to wood monitoring, right? So, it could be useful to discuss about that in the discussion. The same about lines 201-204 and 208-209 (this statement about the good performance of the approach should better go to the discussion after the results are shown and can support that).

Thank you for this important comment, we add a paragraph in the conclusion to explain it as a next step:

"Finally, we think of this work as a first step towards more autonomous systems to detect and quantify wood in rivers. Applying the post-process steps in real time is a realistic next step, because after we extract the correction matrices, which is a time-consuming process, the calculation time for PR, RR enhancement is negligible (less than 0.001s/piece). Moreover, over recent years, automatic visual recognition tasks have progressed very importantly with the advances in machine learning techniques and especially Deep Convolutional Neural Networks (DCNNs) that are now able to answer complex problems in real time. However, our context is very challenging for this class of solution, since wood objects have a highly variable shape, and they are feature in very noisy environments and a high variety of lighting conditions. Most training techniques are supervised, meaning that to train an effective DCNN to solve this problem, we would require an extensive annotated dataset. The solution presented in this work can be used as a first step towards this solution. It can be used to help human operators to quickly build annotated dataset, by correcting its output rather than annotating from scratch. "

The final step related to the wood characterization in terms of size and location is not very clear to me (Lines 210-218). Is the step manual? How the software can select the better view (e.g., when wood is less submerged)?

It was an important point. In fact, in the new version of the postprocessing we first rectify the frames and then calculate the length, so the text in line [226-227] was updated as follow:

"In this step, all images containing the object are transformed from pixel to cartesian coordinates (as will be described in the next section) and the median length is calculated and used as the most representative state. "

Section 2.3 should start by explaining the image rectification, why this is necessary and how it was done, rather than discussing the loss of quality by warping (lines 220-222). Line 222: explain what the particular characteristics are of the image's topology.

There are different rectification approaches (e.g., the methods provided in ArcGIS) that correct image distortion. But huge distortion in streamside videography, due to image warping from foreground to background makes it necessary to apply the 3D rectification process described in this section. Therefore, this sentence in this position was dedicated as a motivation opening for entire section and describes the exceptional topology of our frames (areas of the image farther to the camera provide little detail and are overall very blurry and non-informative.)

Line 223: what other characteristics?

"other characteristics" is removed from the text.

Line 226: add a reference to the toolbox

It is added.

Line 241: a reference is needed here.

It is added.

Line 259: the use of terms like "is intended to work" or "the idea is to load a video" sound like the software is currently not doing that...if so, I think it should be better to describe what the software actually does and discuss in the discussion section the prospects.

Here we described what does the software do at the moment, therefore the text was modified as: "The detection process works as a video file player. The video file (or a stream url) is loaded, ...". As

the software is currently capable of performing these actions, it is a description of the software interface, so we thought it would be better to put it in this position rather than the discussions.

I would structure Section 3 in a different way, following the order in which the work should be done. By describing in the first place the annotation, then learning and finally the automatic detection.

Automatic detection was the main objective of this software. That's why in the software interface the first option is the detection module. Then we used other sections (Annotation, performance, and in the future training module) to enhance detections. Therefore, in an importance hierarchy we first described the main part of the software, and then what we used to improve this module. It should be noted that even without other modules (Annotation and performance) the software can detect objects, but with larger uncertainties.

Lines 278-285: are the different annotation methods all implemented in the software? What is the difference between them? How the use of one influence the results?

The aim of this section was if someone wants to use the software, this section gives a rough idea about different modules, therefore, yes, different annotation methods all implemented in the software. As described in the text we can eighter draw a straight line by selecting endpoints or draw a bounding box around the object. Depending on the study goals different methods can be used, e.g., if the root and branches are also interested, we propose a fully free drawing tool on which one can draw all what he sees in a frame. There are many other features in the software (e.g., masks parameters, connection between pixels, field of view, wood dynamic and static characters etc.) that can be changed based on the propose of study. Evaluating the performance of software under each condition needs a manual book and the aim of this paper were only to present this software and show its general performance.

Line 287: what do the authors mean by "as fine annotation as they wish"? I think the comparison of the annotation methods with other approaches should go in the discussion.

It is substituted by "annotation in different ways, depending on the purpose of the study."

Moreover, what we did here is try to explain what annotations can, and how we decided to use them within this study. Reading the paper back, I think that this choice is not very clear and not well justified. Therefore, the following modification was added to lines 301 to 305:

"This annotation process is time-consuming, so a trade-off must be made regarding the purpose of the annotated database and its required accuracy. Manual annotations are especially important when it is intended to be used within a training procedure, for which different lighting conditions, camera parameters, wood properties, and river hydraulics must be balanced. The rationale for manual annotations in the current study is presented in section **Error! Reference source not found.**"

In a previous line (line 249) four modules are mentioned (detection, annotation, learning and performance), but nothing is explained about learning later. Or is this training?

Sorry for this error, it was "training".

Line 359: this is the first time that wood discharge is mentioned, please provide the units in which is measured only provided in Line 524). It could be useful to define also here wood flux.

Text is modified as: "Ghaffarian et al. (2020), Zhang et al. (2020) show that the wood discharge (m3 per a time interval) can be measured from flux or frequency of wood objects (pieces number per a time interval)."

Table 3: total is the mean?

In fact, it is the condition when we consider all 6 days together. E.g., Qmax: max discharge among all 6 days or Recall rate: the total RR all together.

Table 4: are all the correlations significant? If not, authors could use bold font to highlight the statistically significant ones.

It is done.

Lines 488-496: the use of post process matrices is not very clear to me, and neither the graph shown in Figure 15. In addition, in Figure 15 the x-axe label is cut.

The authors agree with this point and the corresponding figure is removed.

Section 4.3.2. modelling here is unclear, I think authors mean estimating or predicting...

Line 504: model? What type of model? Although explained later, it sounds just like a threshold that is set up based on the manual annotation. This is related to my previous comment about modelling. I think better using estimation.

The term "Model" is changed to "Estimate".

Line 501: the software works or worked well...but, what do the authors mean by well?

The text is changed to "both PR and RR are much higher" instead of "worked well".

Lines 508-509: not clear.

It is supposed that the most realistic length distribution in video monitoring technique can be achieved by annotations of an observer. So, these annotations were used as a benchmark for extrapolating the length distribution of the software detections. Therefore, to explain the idea the sentence was completed by, "and assuming that this line is the most realistic distribution which can be estimated from the video monitoring technique, it was served as the benchmark."

The conclusion section is just a summary of the previous content, while it should provide an outlook and state the most important outcome of your work. Do not simply summarize the work, but instead, interpret your findings at a higher level of abstraction.

In this work we tried to introduce a detection software, its algorithm, interface and post processing, therefore the paper does not have enough discussion points. Therefore, we add a small section at the end of the results that talks about how this could be applied in other sites and what the next steps would be.

Lines 563-571: "The results from the current study were all taken from a single site in which a large database of manual annotations was available for developing the correction procedures. In future applications it is unlikely that such a large database would be available. In such cases it is recommended to first ensure that the images collected are of high quality by following the recommendations in (Ghaffarian et al., 2020; Zhang et al., 2021). As data are collected, the automatic algorithm can be run to identify periods of high wood flux. Manual review of other high-water periods is also recommended to assess whether lighting conditions were preventing the detection of wood.

When suitable flood periods with floating wood are identified, manual annotations should be done to create the correction matrices."

**Further specific comments on Figures:**

Figure 5: is (a) really needed? Not sure about the additional information it conveys.

This figure was necessary to explain the rectification process. However, in the new arrangement we merge it with the rectification network (Fig 10).

Figure 6 to 8; are they really necessary? Figure 6 probably not...and 7 and 8 could be merged into one.

Figure 6 is removed and 7, 8 are merged together.

The number of figures in the manuscript is really large. In order to reduce the number of figures I would suggest the following:

Figure 11 could be merged to Figure 9 (study site and studied period).

It is done

Figure 12 and 13 could also be merged.

It is done

Figure 14: why e and f plots have different area? The same question about k and l.

The reason is: PR = TP / (TP+ FP) and RR = TP / (TP+ FN). So, in one condition if we have only FN, the Matlab code calculates PR = NAN and RR = 0.

Figure 16: using only colors (especially red and green) could be an issue for color blind readers, consider using also different line types.

It is done

Figure 17: use different shapes for the two dot groups.

It is done

---

## Author Comment (AC2) · 12 Mar 2021

First, we want to thank referee #2 for his/her recommendations. Applying them can make this work smoother and more understandable. We corrected all comments in the manuscript. The changes are noted in blue in the revised version of the manuscript.

Overall, I believe the work would benefit from further clarification of the masking procedure and the post-processing steps. Given how important these steps are for accurate analysis, a more detailed description of these components would assist readers when using the software. For example, L151-153,283-285, how are images chosen for the annotation and how many are recommended?

We tried to clarify our explanations in different parts and precisely concerning the masking procedure. Therefore, the following correctios were applied to the text:

Lines 161-163: To set the algorithm parameters, pixelwise annotations of wood under all the observed lighting conditions were used to determine the mean ($\mu$) and standard deviation ($\sigma$) of wood piece pixel intensity.

Lines 301-305: This annotation process is time-consuming, so a trade-off must be made regarding the purpose of the an-notated database and its required accuracy. Manual annotations are especially important when it is in-tended to be used within a training procedure, for which different lighting conditions, camera parameters, wood properties, and river hydraulics must be balanced. The rationale for manual annotations in the cur-rent study is presented in section 5.1.

Lines 563-571: The results from the current study were all taken from a single site in which a large database of manual annotations was available for developing the correction procedures. In future applications it is unlikely that such a large database would be available. In such cases it is recommended to first ensure that the images collected are of high quality by following the recommendations in (Ghaffarian et al., 2020; Zhang et al., 2021). As data are collected, the automatic algorithm can be run to identify periods of high wood flux. Manual review of other high-water periods is also recommended to assess whether lighting conditions were preventing the detection of wood. When suitable flood periods with floating wood are identified, manual annotations should be done to create the correction matrices.

However, it should be noted that in this study we only examine one field site and there was no way of knowing at this point what recommendations would be for another site.

A recommendation or assessment of the amount of training data would assist future users of the software. In the post-processing (L475) it was unclear to me how the precision matrices were applied to the day of interest. Are these matrices generated for each piece of wood detected?

To improve the precision, the eight parameters were calculated for each detected object, these values as four pairs were then used in this figure to calculate the precision for each pair. Finally, the total precision was the average of all four precisions in figure 14. Therefore, these parameters were calculated separately for each detected object. To clarify it in the text we add the following sentence to the text: "Having the value of the eight key parameters (four pairs of parameters in Fig 9) for each piece of wood, …"

What about the assessment of wood length between the manual annotations with the software?

This comment was an important part of the work to clarify how we estimate smaller pieces which software could not detect. Based on our observations the software works well for large pieces (O(10m), figure 14.b, c). Also, in some previous works (e.g., Ghaffarian et al. 2020) we observed that the length distribution is unique in a section of the river. Therefore, the length distribution of annotations was used as a benchmark to estimate smaller wood pieces. therefore, we add the following sentence to line [505]: "Based on this idea, the final step in the post processing is to estimate smaller wood pieces that were not detected by the software using the length distribution extracted by the annotations."

Also, at the times in the manuscript, it was unclear when you mention "it" or "object" what you were actually referring to- the software, a wood piece, the entire video set? I highlight some specific examples of this below.

Concerning the term "object" we used it as a detection which can be a piece of wood (TP) or a false detection (FP). However, it is important to clarify the reference of each term and this comment was taken into account more precisely in the revision of the text.

**Specific comments:**

I may have missed it, but a link to the software or the details of how to access the software would be useful.

The software will be freely available on GitHub soon and it will be indicated in the final version of the paper in section "Code/Data/Sample availability".

L164: I was confused on your use of "stable" here. Wouldn't "stable" wood detections be false positives as they would actually be vegetation or the bridge being detected?

The sentence should be corrected as follows:

Meanwhile, the intensity of pixels that keep on displaying wood tend to be rather stable

L150 and 168: You first define variable x as the pixel light intensity and then again as position in the image. Please revise.

Instead of "x", we use "i".

L429: How did you measure the correlation between these parameters?

To clarify the text the following sentence was added: "… by calculating each one of the eight parameters for all detections as one vector and then calculating the correlation between each pair of parameters."

L443: What is "it"?

"It" substituted by: "This spatial gradient in precision is likely …".

**Technical comments:**

L209: Replace "by" with "be"

It is corrected.

L222: should be "calculate"

It is corrected.

L241: Please revise sentence

It is revised.

L278: Should be "manual"

It is corrected.

L288-290: Some grammar issues, please revise.

It is corrected.

L452: Unclear what you mean by "many noises in a frame".

For clarification it is referred to **Error! Reference source not found.**.

Table 5: Is PR improvement also a percentage?

Yes, (%) is added.

L506: Replace "counting" with "counted"

It is corrected.

Fig 15: Horizontal axis label is cut off

It is corrected.

---

## Author Response (AR2)

The authors thank the referees for their thorough review, and the positive comments made on our work. The comments are addressed below one by one and we hope that the referee will find our answers satisfying. The changes are noted in blue for referee #1 and green for referee #2 in the revised version of the manuscript.

**Comments of the anonymous Referee #1**

Line 240: not clear what topology of the images means here. I would suggest removing "given the topology of our images" and keep the sentence as follows: "Therefore, image rectification was necessary…"

Done

Table 4: please, provide the details about the applied test of correlation (also in Line 433) and explain what the bold values mean (significant correlation?) in the caption.

The correlation test was applied on 8 different vectors, each represents on key parameter. It is described in lines 433 to 435 as: "we first tested the correlation between the factors identified in the previous section by calculating each one of the eight parameters for all detections as one vector and then calculating the correlation between each pair of parameters".

To clarify the table, we add "Values in bold show significant correlation." In caption of the table and "(the bold values)," in line 435.

The same for Figure 10 and the related text (e.g., Line 516)

Thank you for this comment. Here we used the length distributing of TPs and TP+FN as two vectors and calculate the correlation coefficient between them. Therefore, to clarify it, we add "The correlation coefficient between them the length distribution of TP as one vector and TP+FN as the other vector was calculated" to the line 516.

Line 523: virtual? I would simply say that missed pieces were estimated.

The word virtual is removed from the text. Then, for more clarification we add an additional sentence: "…, note that these pieces were imaginary pieces inferred from the wood length distribution and were not detected by the software".

**Comments of the anonymous Referee #2**

L119: What is an example of a larger dataset? Video over 50 minutes? 1 hour?

We add "(i.e., Video segments more than 1hr)." to the text.

Figure 1e: label within figure should be "15 Dec"?

Done

L153: In reading the paper, I was confused when the camera parameters are specifically input in the processing steps. I suggest pointing to the section where these different scales are utilized in the analysis.

It is correct that using this sentence at this stage is a bit confusing, so we remove it from this part.

L192: You mention noise in the image related to water waves, but is vegetation in the image also considered noise here? Please clarify.

We add "or surrounding vegetation" to line 404 after surface water waves.

L323-324: Do you have a sense if annotating for occurrence, objects, or pixels produces more accurate results? Which would you recommend for a first-time user with their own data? To get more wood pieces or more of the same piece over time?

It depends on the scope of the study, for example, if the statistics of wood pieces is important, more pieces result in smoother statistics. If we want to train software (e.g., what happens in AI) or if the dynamics of wood is important for us, it is necessary to trace the trajectory of wood by multiple detections of same pieces. Here, we only wanted to show that the annotation module able to records pieces in different modes.

L405: Change sentence to "Both of these conditions are noted in Figure 8a."

The darker patches are actually visible in all three figures (8, a, b, c) but it is highlighted only in one of them (8, a), and what is mentioned in the text is "Both of these conditions can be seen in **Error! Reference source not found.** which is highlighted in **Error! Reference source not found.**.a." which is correct.